# Probiotic Supplementation in Chronic Kidney Disease: Outcomes on Uremic Toxins, Inflammation, and Vascular Calcification from Experimental and Clinical Models

**DOI:** 10.3390/toxins18010006

**Published:** 2025-12-20

**Authors:** Teresa Obrero, María Victoria Pendón-Ruiz de Mier, Jose E. Gordillo-Arnaud, María José Jiménez Moral, Victoria Vidal, Fátima Guerrero, Andrés Carmona, María Encarnación Rodríguez-Ortiz, Ana Isabel Torralbo, Raquel Ojeda, Cayetana Moyano, Mercedes Sanchez-Ramade, Juan Mesa, Daniel J. López-Ruiz, Karen Valdés-Díaz, Raquel María García-Sáez, Daniel Jurado-Montoya, Cristian Rodelo-Haad, María Álvarez-Benito, Griet Glorieux, Mariano Rodríguez, Sagrario Soriano-Cabrera, Juan Rafael Muñoz-Castañeda

**Affiliations:** 1Research Group: “Calcium Metabolism. Vascular Calcification”, Maimonides Biomedical Research Institute of Cordoba (IMIBIC), Reina Sofia University Hospital, University of Córdoba (Spain), 14004 Cordoba, Spain; 2Nephrology Service, Maimonides Biomedical Research Institute of Cordoba (IMIBIC), Reina Sofia University Hospital, University of Córdoba (Spain), 14004 Cordoba, Spain; 3Redes de Investigación Cooperativa Orientadas a Resultados en Salud (RICORS), Carlos III Health Institute, 28029 Madrid, Spain; 4Research Group: “Nephrology. Cellular Damage in Chronic Inflammation”, Maimonides Biomedical Research Institute of Cordoba (IMIBIC), Reina Sofia University Hospital, University of Córdoba (Spain), 14004 Cordoba, Spain; 5Radiodiagnostics Service, Reina Sofia University Hospital, 14004 Cordoba, Spain; 6Department of Internal Medicine and Pediatrics, Nephrology Division, Ghent University Hospital, 9000 Ghent, Belgium

**Keywords:** chronic kidney disease, probiotics, uremic toxins, inflammation, vascular calcification, microbiota

## Abstract

Chronic kidney disease (CKD) is associated with gut microbiota alterations that contribute to increased inflammation and the generation of uremic toxins and may worsen the disease progression. While probiotics may improve the pro-inflammatory cytokine profile, their effects on mineral metabolism, vascular calcification (VC), and CKD progression remain unclear. We aimed to evaluate the impact of a commercial probiotic (Probimel) supplementation on kidney function, mineral metabolism, inflammation and VC in both an experimental rat model and patients with advanced CKD and VC. The experimental model of VC was performed through 5/6 nephrectomy (Nx), a high-phosphate diet, and calcitriol. Animals were divided into three groups: Sham, Nephrectomy, and Nephrectomy + Probiotic. In the exploratory clinical study, 23 patients with advanced stage 5 CKD and VC were randomized and either received or did not receive daily probiotics for 6 months. Kidney function, mineral metabolism, uremic toxins, inflammation, VC, and fecal microbiota were evaluated. Probiotic supplementation decreased interleukin-6 (IL-6) and interpheron-γ (IFN-γ) and levels of the uremic toxin, indoxyl sulfate (IS), in the experimental model. However, no clear evidence of improvement in kidney function or vascular calcification was observed in either rats or patients with this probiotic. Under our experimental and clinical conditions, the selected probiotic did not modify key parameters related to CKD progression or VC.

## 1. Introduction

Chronic kidney disease (CKD) has a high global incidence and prevalence with an upward trend, currently ranking as the seventh leading risk factor for mortality worldwide [1]. CKD patients experience elevated oxidative stress, chronic low-grade inflammation, and accumulation of uremic toxins [2,3]. These changes contribute to endothelial dysfunction, cellular senescence, and vascular calcification (VC), with the subsequent development of cardiovascular diseases (CVDs) and the associated increase in morbidity and mortality [4,5,6].

VC is an active process that begins in the arterial medial layer and gradually extends to the intima. This process is primarily driven by vascular smooth muscle cells (VSMCs) that, under pathological conditions, undergo phenotypical transdifferentiation into osteoblast-like cells. This transformation is predominantly induced by elevated extracellular phosphate (P) levels, a common finding in CKD [7,8]. Currently, treatment options for patients with advanced CKD and VC are implemented when the patient has kidney failure.

Alterations in gut microbial homeostasis and the generation of metabolites associated with dysbiosis may contribute to kidney damage, ultimately promoting renal fibrosis. [9,10]. It is well established that in patients with CKD, the uremic state, dietary protein restriction, and polypharmacy lead to gut microbiota dysbiosis and impaired intestinal barrier function [11]. This disruption in microbial composition exacerbates the generation of uremic toxins such as indoxyl sulfate (IS), p-cresyl sulfate (pCS), and trimethylamine N-oxide (TMAO) by the overgrowth of proteolytic bacteria [12]. Changes in microbiota are also associated with increased inflammation and intestinal permeability, facilitating the translocation of lipopolysaccharides (LPS) and other bacterial components into the bloodstream, unleashing chronic low-grade inflammation. This inflammatory state accelerates the phenotypic change of VSMCs to osteoblastic cells, thus pushing through VC [13]. In addition, changes in the microbiota also reduce the beneficial short-chain fatty acids (SCFAs), such as acetate, propionate, and butyrate, that are critical to maintain intestinal and vascular health [14,15,16,17].

SCFAs produced by beneficial bacteria, such as *Ruminococcaceae* and *Lachnospiraceae,* have been shown to exert anti-inflammatory effects and regulate calcium-phosphate homeostasis, potentially mitigating the progression of VC [18]. Conversely, dysbiosis and increased intestinal permeability can accelerate VC by allowing pro-inflammatory microbial metabolites to enter circulation and negatively affect vascular health [19,20,21].

Probiotics have emerged as a cost-effective and safe alternative for treating numerous chronic conditions [22]. In the context of CKD, probiotics supplementation has been evidenced to modulate the microbiota composition, reducing uremic toxin production and inflammation [23,24,25]. Additionally, it has also been demonstrated that specific probiotics, inducing changes in gut microbiota, can modify P absorption, reducing serum P levels [26]. However, it remains to be explored whether probiotic supplementation may help to mitigate the development of VC. Thus, there is growing interest in evaluating the potential role of gut microbiota and probiotics to mitigate the adverse effects of high P, resulting in slowing down the progression of CKD.

Our study aimed to explore through experimental and clinical studies the impact of probiotic supplementation on kidney function, mineral metabolism, inflammation, uremic toxins, and VC in CKD.

## 2. Results

### 2.1. Experimental Model of Vascular Calcification in Rats

#### 2.1.1. Changes in Kidney Function and Mineral Metabolism Parameters

As expected, all kidney function parameters were significantly altered in the 5/6 nephrectomy (Nx) group compared to the simulated (Sham) group (Table 1). Thus, plasma phosphate (P) and creatinine (Cr), fractional excretion of phosphate (FEP), end-proximal tubule phosphate concentration (ePTpc70), and intact fibroblast growth factor 23 (iFGF23) were significantly elevated in the Nx group as compared to Sham rats. However, probiotic administration did not affect the kidney function or mineral metabolism parameters (Table 1).

#### 2.1.2. Inflammation in the Rat Experimental Model

We performed a Milliplex assay to analyze five inflammatory parameters, namely interleukin-6, (IL-6), interpheron- γ (IFN-γ), tumor necrosis factor-α (TNF-α), interleukin-10 (IL-10), and interleukin-1β (IL-1β) in rats. We observed that three of the cytokines studied, IL-6, IFN-γ, and TNF-α (Figure 1A–C), showed a significant increase in the Nx rat group compared with the Sham rats, and a tendency toward an increase in IL-10 (Figure 1D). Furthermore, IL-6 and IFN-γ showed a significant decrease in Nx rats after probiotic administration, while TNF-α and IL-10 exhibited a decreasing trend. No changes were observed in IL-1β levels (Figure 1E). These results suggest that probiotic administration, in this case, may help reduce the microinflammatory state associated with uremia.

#### 2.1.3. Vascular Calcification in the Rat Experimental Model

To evaluate VC in rats, micro-computed tomography (micro-CT) was performed across the experimental groups. As shown in Figure 2A, Nx rats, with or without probiotic treatment, exhibited massive aortic calcification (white arrows) without a difference between both groups. Similar findings were observed when Von Kossa staining was analyzed (Figure 2B), which revealed extensive calcium (Ca) deposits in the medial layer of the thoracic aorta in both Nx groups. Additionally, uremic rats showed a significant increase in Ca mineral content in the thoracic aorta (Figure 2C) compared to sham rats. No significant differences were observed between the uremic rats treated with probiotics and the Nx group.

#### 2.1.4. Quantification of Uremic Toxins in the Rat Experimental Model

Plasma concentrations of the total fraction of uremic toxins, including pCS, p-cresyl glucuronide (pCG), and IS, were significantly elevated in Nx rats compared to the sham-operated group (*p* < 0.05). No statistically significant differences were observed between the Nx group and the group receiving probiotics (Nx + Prob). Nonetheless, a trend was noted, with lower levels of IS and higher indole-3-acetic acid (IAA) concentrations in the Nx + Prob group (Figure 3). The free fraction of these uremic toxins in plasma remained below the limit of detection in all experimental groups.

#### 2.1.5. Characterizing Fecal Bacterial Diversity in the Rat Experimental Model

In the alpha diversity analysis, which refers to the diversity within a single sample or community, no significant differences were observed among the groups in terms of richness, defined as the total number of distinct taxonomic groups present in a sample (observed Operational Taxonomic Units, OTUs) (Figure 4A) or evenness, referring to the relative difference in abundance of individuals within each taxonomic group in a sample (Figure 4B). Similarly, unsupervised principal component analysis (PCA) (Figure 4C) did not reveal differences in fecal microbial composition between the groups, indicating no variation in beta diversity.

In the analysis of the taxonomic profile, at the phylum level, Firmicutes, Verrucomicrobiota, and Bacteroidota represented more than 95% of the gene sequences. The remaining 5% was mainly composed of Actinobacteriota, Proteobacteria, Deferribacterota and Desulfobacterota among others (Figure 5A). At the family level, significant differences were identified between Sham and Nx groups. Specifically, the Nx group showed a marked increase in Lachnospiraceae (** *p* < 0.01 Nx vs. Sham), together with a significant decrease in Rikenellaceae and Muribaculaceae (* *p* < 0.05 and ** *p* < 0.01 Nx vs. Sham, Figure 5B). At genus level, the Nx group exhibited a significant rise in *Lachnoclostridium*, *Blautia*, *Sellimonas*, *Eubacterium*_*nodatum*_group, *Marvinbryantia*, and *Ruminococcus*_*torques*_group. In contrast, the genera *Muribaculaceae*, *Oscilibacter*, *Rikenellaceae*_RC9_gut_group and *Lachnospiraceae*_NK4A136_group experienced a notably reduction in abundance compared to the Sham group (* *p* < 0.05 and ** *p* < 0.01 vs. Sham, Figure 5C). At species level, the Nx group exhibited a significant reduction of *Lactobacillus reuteri*, with a concurrent increase in the abundance of *Clostridium*_*scindens* compared to the Sham group (* *p* < 0.05 vs. Sham, Figure 5D).

Following probiotic administration, no significant differences at the phylum level were found. However, the relative abundance of several bacterial populations decreased compared to the Nx group, specifically, bacteria from the Erysipelotrichaceae family and genera such as *Faecalitalea* and *Blautia*. Other genera, such as *Erysipelatoclostridium* and *Sellimonas*, were drastically reduced or disappeared. The species *Bacteroides dorei* also had lower abundance in the Nx + Prob group as compared to the Nx group (# *p* < 0.05, ## *p* < 0.01 vs. Nx group, Figure 5B–D).

Using the bacterial taxa that exhibited significant differences between the groups, a Linear Discriminant Analysis (LDA) was performed. As shown in Figure 5E, the Sham group is enriched with the genera *Rikenellaceae RC9* gut group and *Lachnospiraceae NK4A136* group, while the Nx group was predominantly characterized by the genera *Sellimonas* and *Oscillibacter*. Additionally, in the Nx group, there was an enrichment of the genera *Erysipelatoclostridium*, *Sellimonas*, and *Oscillospiraceae* as compared to the Nx + Prob group.

We performed a correlation analysis (Figure 6), which clearly illustrates the relationships between bacterial taxa and parameters related to VC and mineral metabolism. It is highlighted that the Rikenellaceae family exhibited a negative correlation with plasma creatinine, iFGF23, and ePTpc70 levels. Within this family, the genus *Alistipes* also showed negative correlations with these three parameters, as well as with Ca levels in the thoracic aorta. Additionally, the genera *Clostridium sensu stricto 1* and *Anaerotruncus* also correlated negatively with Ca in the thoracic aorta.

### 2.2. Clinical Study: Results

#### 2.2.1. Kidney Function and Mineral Metabolism Parameters in Patients with CKD and VC

The analysis of demographic and clinical characteristics between the groups with VC who were or were not administered probiotics revealed no statistically significant differences at baseline. Thus, the groups were comparable in terms of kidney function, age, gender, body mass index (BMI), blood pressure (BP), or pulse pressure (PP). Additionally, there were no significant differences in the prevalence of hypertension (HT), diabetes (DM), dyslipidemia (DL), hyperuricemia (HU), smoking status, ischemic heart disease (IHD), cardiovascular disease (CVD), advanced CKD or the proportion of patients undergoing dialysis. Specifically, in the VC group (n = 11), three patients were receiving peritoneal dialysis (PD) and two were undergoing hemodialysis (HD). In the VC + Prob group (n = 6), one patient was under HD treatment. (Table 2).

After six months of probiotic administration, no significant changes were observed in kidney function, as evidenced by the serum creatinine (Cr) or eGFR. Similarly, markers of mineral metabolism, such as PTH, iFGF23, Vitamin D (1,25OH and 25OH), magnesium (Mg), Ca, or P, remained unchanged. In addition, no differences were found either in iron or ferritin levels (Table 3).

Nutritional status was stable throughout the study. Specifically, there were no changes in BMI (Table 2). Furthermore, there are no changes in dietary patterns over time or between groups in phosphate, calcium, or protein, nor in processed foods or soft drinks consumed in the diet (Table 4).

#### 2.2.2. Inflammation Profile in Patients with CKD and VC

Ninety-two inflammation-related proteins were analyzed using the Olink Target 96 Inflammation panel (Appendix A). Paired analyses revealed no significant differences in any of the parameters assessed, either at baseline or after 6 months of probiotic administration.

#### 2.2.3. Vascular Calcification in Patients with CKD and VC

As shown in Figure 7, after probiotic administration, no significant changes were observed in the scores, Kauppila (Figure 7A) and Adragao (Figure 7B), (Kauppila index was 8 vs. 9.6 and Adragao index was 3.8 vs. 1.3 in VC and VC + prob groups, respectively). These results were also confirmed by non-contrast micro-CT. No significant differences were detected in the Ca volume (C) or mass (D) quantified by CT in the L1–L5 vertebrae.

#### 2.2.4. Quantification of Uremic Toxins in Patients with CKD and VC

Plasma concentrations of the total fraction of pCS, pCG, IS, and IAA (Figure 8A–D) remained stable over the six months follow-up period, regardless of probiotic administration. Similarly, no differences were observed in the concentrations of their free fractions (Figure 8E–H).

#### 2.2.5. Characterization of Fecal Bacterial Diversity in Patients

No significant differences were observed in alpha diversity, either in terms of richness (Figure 9A) or evenness (Figure 9B). However, beta diversity analysis revealed significant differences in the baseline group at the end of the follow-up, showing a distinct microbiota composition (*p* = 0.004) (Figure 9C).

In this human cohort, Firmicutes, Bacteroidota, Verrucomicrobiota, Actinobacteriota and Proteobacteria were the most abundant phyla, representing more than 97% of the gene sequences. The remaining 3% was composed of Deferribacterota, Desulfobacterota, Campilobacterota and Fusobacteriota. Firmicutes was the most abundant phylum, followed by Bacteroidota in all groups. Figure 10A shows that with respect to baseline, in the 6-month follow-up group (without Probiotic) there was a significant decrease in Firmicutes (** *p* < 0.01) and an increase in Bacteroidota (* *p* < 0.05). In the group VC + Prob there was also a significant decrease in the abundance of Firmicutes compared to baseline. At the end of the follow-up period, the probiotic administration did not result in significant differences at the phylum level as compared with the baseline. When changes at the family level after six months follow up were analyzed, patients with VC showed an increase in the abundance of the families such as Bacteroidaceae, Rikenellaceae, Tannerellaceae, and Christensenellaceae and a decrease in Lachnospiraceae family (* *p* < 0.05, Figure 10B). Likewise, patients that received probiotics also had increased abundance of Rikenellaneae, but not the other families that were found altered in those patients who did not receive treatment. As expected, patients administered the probiotic showed a significant increase in the Ruminococcaceae family, being one of the families present in the probiotic.

At genus level, it is interesting to highlight that after the follow-up period, genera such as *Bacteroides*, *Alistipes*, *Parabacteroides*, and *Christensenellaceae_R-7*_group increased their relative abundance compared to baseline in the VC group, while *Ruminococcus torques* group genera were significantly decreased (* *p* < 0.05). These changes were not statistically significant in the VC + Prob group. However, when comparing the group that received probiotics with the group that did not at month 6, it was shown that both genera-CAG-352 and *Faecalibacterium* were significantly increased (# *p* < 0.05 vs. VC Month 6, Figure 10C).

Regarding the analysis of species, it was found that *Bacteroides uniformis* and *Alistipes onderdonkii* were significantly more abundant in VC at 6-month than at baseline. These species were not significantly different in the patients receiving probiotics for 6 months. In addition, when both groups were compared at 6 months, there was a significant increase in the abundance of an unidentified species from the genus *Faecalibacterium* (# *p* < 0.05 vs. VC Month 6 group, Figure 10D).

## 3. Discussion

This work investigated the influence of probiotic administration on VC in both preclinical and clinical settings, with particular attention to changes in kidney function, mineral balance, uremic toxins, and systemic inflammation.

The probiotic used (Probimel) was chosen pragmatically as a commercially available product containing Lactobacillus and Ruminococcaceae. These strains have been previously reported by others in the literature to potentially modulate gut microbiota, reduce uremic toxins generation, and influence mineral metabolism [27,28,29]. The dose (10^9^ CFU/mL) and the timing (45 days in rodents; 6 months in humans) were selected in accordance with trials available in the literature, such as Mardhatillah Sariyanti et al., who used a dose of 1.5 × 10^8^ or 1.5 × 10^9^ CFU/mL/day in rats for 21 days [30]. I-Kuan Wang et al. administered a dose of 10^7–9^ CFU/day for 42 days in rats and 2.5 × 10^9^ CFU/day for 6 months in patients [31]. Also, Na Tian et al. indicate that it is common to use an administration period of 2 to 6 months in clinical trials [32]. Shaun Sabico et al. used a dose of 2.5 × 10^9^ CFU for 6 months [33].

In the preclinical arm, no significant changes were observed in kidney function parameters (plasma Cr and FEP) or in mineral metabolism markers (iFGF23, Mg, Ca, and P).

VC was assessed by micro-CT, Von Kossa staining and determination of aortic Ca content, showing no significant differences among Nx rat groups, regardless of probiotic administration. In experimental rat models with Nx, recent evidence indicates that the use of certain probiotics not only does not produce benefits in existing VC but may even aggravate it. Wei et al. demonstrated that in an acute experimental model of VC and CKD, the use of probiotics significantly increased the release of extracellular vesicles with the subsequent overactivation of PI3K/Akt pathway and augmented VC [34].

Our research shows that Nx leads to a significant rise in plasma levels of protein-bound uremic toxins such as pCS, pCG, IS, and IAA, aligning with what previous studies have found in CKD models [35]. After administering the probiotic, a trend of decreasing IS levels (*p* = 0.05) compared to the Nx group was observed while pCS, pCG, and IAA levels remained unaltered. This partial effect might be due to the fact that in the probiotic group, fewer bacteria produce indole, an IS precursor, such as those of the genus Oscillibacter, or the probiotic might inhibit the indol-producing capacity of the gut microbiota.

It is interesting to note that these metabolites (IS, IAA, pCS, and pCG) act as endogenous ligands of the aryl hydrocarbon receptor (AhR), which controls the fibrotic and inflammatory pathway in chronic kidney disease. By altering tryptophan metabolism, some Lactobacillus strains can reduce AhR agonist ligands and stimulate the production of beneficial indole derivatives with anti-inflammatory and antioxidant properties, such as indole-3-propionic acid [36,37,38,39]. Therefore, a qualitative alteration of AhR-related metabolites cannot be ruled out, although our data did not demonstrate a significant quantitative reduction in all uremic toxins. Future research should investigate this pathway, assessing AhR activation and the expression of its downstream targets in renal tissue.

Regarding inflammation, we observed a significant decrease in IL-6 and IFN-γ levels following probiotic administration in the Nx rats, suggesting an anti-inflammatory effect under conditions of kidney injury. Our findings are consistent with previous studies in animal models of kidney disease, where probiotics have been shown to reduce pro-inflammatory cytokines such as IL-6, IFN-γ, TNF-α, and IL-18, while improving renal function and decreasing inflammatory cell infiltration [31,40,41]. These results support the hypothesis that probiotics exert protective effects through gut microbiota modulation, reduction in endotoxin translocation, and enhancement of intestinal barrier integrity, thereby contributing to the attenuation of systemic and renal inflammation.

Recent studies have reported kidney-protective effects of specific Lactobacillus strains, such as *Lactobacillus johnsonii*, particularly in models with moderate renal damage and shorter disease evolution [42]. In contrast, our study employed a 5/6 nephrectomy model characterized by advanced and irreversible renal injury, chronic systemic inflammation, and established vascular calcification. Although probiotic supplementation reduced inflammatory markers and tended to lower IS levels, these effects did not translate into improvements in renal function or VC. This suggests that, in advanced CKD, the profound inflammatory and pro-calcifying milieu may limit the therapeutic impact of probiotic modulation. Future studies should evaluate interventions at earlier disease stages and explore targeted formulations to enhance clinical benefit.

In our exploratory clinical trial, all participants were stage 5 CKD patients, some under hemodialysis (HD) or peritoneal dialysis (PD). In these patients, VC was assessed by both Kauppila and Adragao indices, along with Ca volume and mass quantification from non-contrast CT. The use of probiotics in the context of VC remains relatively unknown. However, some studies have suggested the potential role of probiotics in controlling hyperphosphatemia in patients with kidney disease [26,43]. Given that P is a key driver in the development of VC, this has led to considering probiotics as a potential P binder-like therapy. In this regard, probiotics could potentially help to slow down or mitigate the onset of VC by reducing serum P levels. However, our results reveal that 6 months of supplementation with the currently tested probiotic in these patients with CKD and VC are insufficient to modify VC and improve kidney function.

Moreover, probiotic administration was effective in reducing uremic toxins in our experimental model of vascular calcification; however, it failed to do so in the clinical study. This discrepancy may be explained by the pathophysiological differences between advanced CKD and the experimental models. In end-stage renal disease, renal clearance of protein-bound solutes is profoundly impaired, and even a substantial reduction in their microbial generation may not be sufficient to measurably decrease their plasma concentrations [44,45]. In contrast, Nx rats retain residual renal function and intestinal excretory capacity, allowing the reduction in microbial production to be reflected in plasma levels.

If we focus on inflammation, no significant differences were observed in any of the 92 cytokines studied by the PEA technique after the follow-up period. On one hand, some studies support the notion that probiotics may help reduce inflammation [46,47]. On the other hand, our findings are aligned with other studies showing that probiotic supplementation is not always effective in reducing inflammation in patients with CKD in predialysis or dialysis [48,49,50]. The high pro-inflammatory state characteristic of patients with CKD may explain the limited effect of the probiotic on the inflammatory status. The administration of a probiotic like this might be insufficient to elicit measurable beneficial effects within the time frame and at the dose evaluated in this study.

Concerning microbial composition, LDA of the experimental model showed that in the Nx group, there was a loss of bacteria from the *Rikenellaceae* RC9 gut group and *Lachnospiraceae* NK4A136 group, which are associated with beneficial health effects. The *Rikenellaceae* RC9 gut group is known for its role in carbohydrate metabolism, energy production, and butyrate synthesis, a SCFA with various health benefits, including immunomodulatory, anti-inflammatory, intestinal barrier protective, and antioxidant properties [51,52]. The *Lachnospiraceae* NK4A136 group is also known for its ability to ferment plant polysaccharides into SCFAs and ethanol. This group has been described as aiding in correcting dysbiosis and maintaining physiological homeostasis [53]. By contrast, abundant bacteria from the *Sellimonas* and *Oscillibacter* genera, considered harmful, were found in Nx rat. These genera enriched in the uremic rat group have been associated with the progression of various diseases, such as depression, breast cancer, and gout in the case of *Sellimonas* [54,55], and stroke in the case of *Oscillibacter*, which also increases intestinal permeability, promoting inflammation [56,57].

The LDA between the Nx and Nx + Prob groups revealed that in the Nx group there is an enrichment of bacteria from the *Erysipelatoclostridium* genus. Despite the limited number of studies on this genus, it is regarded as an opportunistic pathogen and has been reported to be more prevalent in colorectal adenomas and in individuals with CKD [58,59]. These findings suggest that probiotics might balance the microbiota by reducing the abundance of bacteria that impact negatively on health.

Lastly, a correlation analysis suggested a potential relationship between VC-related parameters and specific bacterial families and genera. A significant negative relationship was shown between the genera *Clostridium* sensu stricto 1, *Anaerotruncus*, and *Alistipes* with the Ca content in the thoracic aorta. *Alistipes* also exhibited a negative correlation with plasma levels of creatinine, iFGF23, PTH, and ePTpc70. These results suggest that a higher abundance of these genera could be associated with a lower mineral burden in the thoracic aorta, highlighting the potential importance of certain bacterial taxa in modulating metabolic and vascular disturbances in the context of CKD. *Alistipes* may have protective effects against diseases, such as colitis, autism spectrum disorders, and various fibrotic hepatic and cardiovascular disorders. However, it has also been associated with the development of anxiety, encephalomyelitis, chronic fatigue syndrome, and depression [60]. *Anaerotruncus colihominis*, the only known species of the genus *Anaerotruncus*, produces SCFAs like acetate and butyrate, and it is suggested to play a role in the dysbiosis observed in CKD patients [61,62]. Similarly, *Clostridium* sensu stricto 1 is another producer of SCFAs, but it has also been positively correlated with p-cresol and associated with CKD progression [63]. There is limited research on the relationship between these bacterial genera and VC; therefore, further investigation may provide valuable insight. Exploring whether these bacteria play a role in the calcification process could determine their potential as markers or therapeutic targets in the context of VC.

Regarding the microbiota in patients, at phylum level, Firmicutes and Bacteroidota were the dominant bacteria across all groups, which was consistent with previous studies on human gut microbiota [64]. However, after the follow-up period, patients without probiotic supplementation exhibited a significant decrease in Firmicutes and a corresponding increase in Bacteroidota. This shift in the Firmicutes/Bacteroidota (F/B) ratio (from 8.42 to 2.82) is notable, as it has been linked to various metabolic and inflammatory disorders [65]. An elevated F/B ratio is a marker of microbiota dysregulation, often associated with increased intestinal permeability and a compromised integrity of the gut barrier [66]. The observed pattern at the phylum level is consistently replicated across all taxonomic categories studied, including family, genus, and species. There was a marked decrease in bacteria belonging to the phylum Firmicutes, which is known to include several uremic toxin-producing bacteria. In contrast, bacteria from the phylum Bacteroidota were increased. This shift in microbial composition, particularly the reduction of Firmicutes, may contribute to a decrease in the production of harmful metabolites. At the same time, the rise in Bacteroidota could be associated with improved gut health and metabolic regulation [67].

In the probiotic-treated group, a significant increase was noted in the Ruminococcaceae family, present in the probiotic, as well as in the CAG-352 and *Faecalibacterium* genera. The *Ruminococcaceae* family is a major producer of SCFAs [68]. *Faecalibacterium*, in particular, is known for its anti-inflammatory role and its ability to produce butyrate, an SCFA with multiple benefits for gut health [69]. These results suggest that probiotics may have a selective effect in modulating the gut microbiome, promoting the growth of beneficial taxa that in turn might contribute to the stability and functionality of the microbiota in CKD patients with VC. Although probiotic administration induced significant shifts in the *Ruminococcaceae* family, implicated in uremic toxin metabolism, these changes did not reflect global alterations in overall community diversity or composition, but rather targeted functional modifications within specific bacterial taxa. This behavior had been observed previously, as described in the review by Nadja B Kristensen and colleagues [70], where they observed how probiotic interventions in healthy adults did not show consistent effects on global diversity indices, although they did show specific changes in taxa.

This study has some important limitations that should be taken into consideration when interpreting the results. The sample size was relatively small, which may influence the extrapolation of these results to larger populations. The small study population was due to restrictive inclusion criteria, along with the concurrence of the recruitment period with the COVID-19 pandemic. For these reasons, our clinical trial was designed as an exploratory pilot study to evaluate the potential effects of probiotic supplementation (Probimel) in patients with advanced CKD. Given the small sample size and the advanced stage of disease in our participants (all in stage 5 CKD), the study lacked sufficient statistical power to detect clinically important differences, but rather to provide preliminary information on safety, feasibility, and potential biological effects, particularly on uremic toxin levels and VC. Although the study has a small sample size, the paired statistical analysis, in which each patient serves as their own control, allows us to conclude that under the specific conditions of our study, six months of probiotic supplementation does not modify renal function or the degree of vascular calcification. Also, our results would be much more significant with the quantification of SCFAs, such as acetate, propionate and butyrate; the absence of functional data limits our ability to infer whether the observed taxonomic changes translated into significant metabolic alterations. Future studies should combine taxonomic and functional profiling.

Additionally, it is important to highlight that, while rodent models, particularly rats, are invaluable for studying microbiota–host interactions under standardized and controlled conditions, significant differences exist between the basal microbiota composition and colonization dynamics of rodents and humans. These differences arise due to host-specific factors, including anatomy, genetics, dietary habits, and lifestyles. Comparative studies have shown that, although some phyla such as Firmicutes and Bacteroidetes dominate in both species, relative abundances, genus and species distributions, and metabolic profiles differ markedly [71]. Thus, while rat models offer mechanistic insight, direct extrapolation to human bacterial load and microbiome responses should be approached with caution [72].

Another potential limitation is the probiotics that we used in the studies. It remains uncertain whether the probiotic formulation in this study is more or less effective than other commercially available options. The wide diversity in probiotic formulations, administration protocols, and dosages poses significant challenges for standardizing outcomes related to probiotic supplementation. Further studies are needed to determine whether alternative combinations of probiotics and prebiotics, longer treatment duration, or earlier intervention in CKD and VC could pave the way for the use of probiotics as a therapeutic tool. Despite limitations related to the small sample size and patient heterogeneity, our results clearly indicate that Probimel was ineffective in reducing or halting VC in our settings.

## 4. Conclusions

In conclusion, our results, obtained from both experimental models and patients with CKD, reveal that probiotic supplementation does not seem to affect VC. Nevertheless, in the experimental model, we observed a reduction in circulating pro-inflammatory cytokines (IL-6 and IFN-γ) as well as a trend toward in the uremic toxin, IS, levels, supporting a potential anti-inflammatory and detoxifying effect under controlled conditions. By contrast, the exploratory clinical study showed no significant improvement in kidney function, inflammation, uremic toxin generation, or mineral metabolism following probiotic administration. However, a potential trend suggests that probiotics may induce changes in microbiota in a direction opposite to that observed in patients with CKD and VC who did not receive probiotic supplementation. Therefore, we conclude that, in the short term, the benefit of probiotics on VC and kidney function appears to be of limited clinical relevance. The potential benefit of probiotics may need to be explored over a longer period, or rather as a preventive strategy in patients with a lower degree of VC, less advanced kidney impairment, and dysbiosis, or alternatively with different probiotic formulations selected based on in vitro proof of concept.

## 5. Materials and Methods

### 5.1. Experimental Study

Animal experiments were conducted at the animal facility of the Maimonides Biomedical Research Institute in Córdoba (IMIBIC). Male Wistar rats (Charles River Laboratories, Wilmington, MA, USA), approximately 9 weeks old and weighing around 270 g, were housed individually under controlled temperature, humidity, and a 12 h light/dark cycle. They had ad libitum access to food (Altromin GmbH, Lage, Germany) and water. The study was approved by the Ethics Committee for Animal Research at the University of Cordoba (file number: 08/03/2021/031). It was confirmed that all rats were visibly healthy before the study, and no exclusion criteria were established in any experimental group. The animals received humane care in compliance with the Principles of Laboratory Animal Care formulated by the National Society for Medical Research and the 2010/63/EU Directive. The study is reported under ARRIVE guidelines.

Figure 11 illustrates the experimental protocol. Uremia was induced by Nx, following the protocol established in previous studies by our group [27]. After the second surgery, the P content of the diet was 0.9%, and calcitriol was administered at a dose of 25 ng/kg/48 h i.p. for 30 days. Subsequently, for the following 15 days, the dietary P content was increased to 1.2%, and calcitriol (Kern Pharma S.L., Terrasa, Barcelona, Spain) was injected at 40 ng/kg/48 h.

The animals were randomly assigned to the different experimental groups without applying any specific stratification or selection criteria during the randomization process. Rats were allocated into three groups: Sham (saline by oral gavage, n = 9), Nx (saline by oral gavage, n = 14) and Nx + Probiotics (by oral gavage, n = 6). The animals were consecutively numbered and labeled with color-coded tags specific to each group to prevent potential confusion during measurements and procedures. In addition, all authors involved in the experimental development were aware of the distribution of the experimental groups. The probiotic selected was a commercially available formulation obtained from Probimel (Laboratorios Biotecnológicos Probimel S.L., Seville, Spain). It consisted of a liquid food supplement based on fermented milk (skimmed milk powder, 0.05 g, dissolved in 4.95 mL of water) containing 10^9^ colony-forming units (CFU)/mL of live probiotics, a mixture of species belonging to the Lactobacillus and Ruminococcaceae families. Specifically, the strain used was Lactobacillus acidophilus DMG017 (Probimel, Laboratorios Biotecnológicos Probimel S.L., Seville, Spain). Animals received a daily oral gavage of either saline or probiotic solution (0.5 mL/day) for a total of 45 days. The day before the sacrifice, all rats were placed in sterile metabolic cages to collect 24 h urine and feces. After 45 days, just before the sacrifice, rats underwent micro-computed tomography (micro-CT) to observe vascular calcified tissue using a Skyscan 1176 (BrukermicroCT, Kontich, Belgium). Subsequently, rats were sacrificed by aortic puncture and exsanguinated under general anesthesia Isoflutek 1000 mg/g (Laboratorios Karizoo S.A., Barcelona, Spain). Blood was collected in heparinized syringes, and plasma was separated by centrifugation and stored at −80 °C until analysis. In addition, two rings from the thoracic aorta were collected and fixed in 4% formalin (AppliChem GmbH, Darmstadt, Germany), and the rest of the aorta was stored at −80 °C. After the experimental procedure, no animals were excluded from the study.

#### 5.1.1. Biochemical Measurements

In plasma, an ELISA kit was employed to quantify the levels of intact Fibroblast Growth Factor-23 (iFGF23, Kainos Laboratories, Tokyo, Japan). Phosphate and creatinine levels were measured in plasma and urine by spectrophotometry with commercially available kits from BioSystems SA (Barcelona, Spain).

#### 5.1.2. Determination of Cytokines

Plasma cytokines (IL-1β, IL-6, IL-10, TNF-α and IFNγ) were measured by using the MILLIPLEX^®^ MAP Rat Cytokine/Chemokine Magnetic Bead Panel (MercK KGaA, Darmstadt, Germany; Ref. RCYIL1B-MAG, RCYIL6-MAG, RIL10-MAG, MCYTNFA-MAG and RINFG-MAG, respectively; Millipore, MA, USA) following the manufacturer’s instructions. Data analysis was performed using the Bio-Plex^®^ 200 System.

#### 5.1.3. Histology

Fresh aortic tissue was fixed in 4% formalin, embedded in paraffin and cut into 3 μm sections. To determine the level of VC, Von Kossa staining in paraffin-embedded aorta sections was performed to detect and quantify VC. Ca deposits were visualized as brown.

#### 5.1.4. Quantification of Calcium Content from Thoracic Aortas

Aorta segments were demineralized in 10% formic acid, and the calcium content in arterial tissue was measured from the supernatant by spectrophotometry (BioSystems S.A, Barcelona, Spain), according to previous studies from our group [73].

#### 5.1.5. Gut Microbiome Analysis in the Rat Experimental Model and in Patients from the Clinical Study

The composition and structure of the fecal microbiota were evaluated through amplification and sequencing of the V3–V4 regions of the 16S ribosomal RNA gene. For this purpose, fecal samples preserved at −80 °C were sent to Microomics Systems S.L. (Barcelona, Spain), which carried out the entire process of extraction, amplification, sequencing, and preliminary statistical analysis. Briefly, DNA was extracted with commercial kits, and amplification was performed using 25 PCR cycles, with quality control measures that included both positive and negative controls. The positive control consisted of a Mock Community, processed identically to the samples. Sequencing was conducted on an Illumina MiSeq platform (Illumina Inc., San Diego, CA, USA) (2 × 300 bp) to generate the libraries.

The raw, un-demultiplexed reads (forward and reverse) were processed using QIIME2. The DADA2 software was employed to carry out several steps: trimming of reads (5′ adapters were removed according to the primer length used plus 2 nucleotides), quality filtering (3′ ends were trimmed to remove low-quality bases), denoising, and read-pair merging (forward and reverse reads were merged to reconstruct the amplicons), as well as phylotype identification (biological entities were inferred from sequence data using the DADA2 pipeline).

Phylotype data were used to calculate alpha diversity richness and evenness. Richness, or observed operational taxonomic units (OTUs), was defined as the number of distinct phylotypes present in a community. Evenness was calculated using Pielou’s index, quantifying the numerical uniformity of the community in terms of phylotype number and abundance. Alpha diversity comparisons were conducted using a Generalized Linear Model (GLM). Richness was analyzed with the R package MASS v.7.3-54, while evenness was evaluated using the glmmTMB v.1.1.8 package. When a Generalized Linear Mixed Model (GLMM) was applied, richness was analyzed using the NBZIMM v.1.0 package, and evenness using betareg v.3.1-4. The significance threshold was set at 0.05.

Phylogenetic distances between OTUs were assessed using Mafft and FastTree. OTU and phylogenetic data were used to compute the beta diversity metric Unweighted UniFrac, which incorporates phylogenetic information and compares samples based solely on the presence or absence of OTUs. Beta diversity distance matrices were used to perform Principal Coordinates Analysis (PCoA) and to generate ordination plots using R version 4.2.0. Statistical significance between groups was evaluated using PERMANOVA and ANOSIM tests, while the PERMDISP test was applied to identify location versus dispersion effects. The significance threshold was also set at 0.05.

Taxonomic assignment of phylotypes was carried out using a Bayesian Classifier trained with the Silva database version 138 (99% OTU sequences, full length). Differential taxon abundance was assessed using Generalized Linear Models with a negative binomial distribution, applying either the MASS v.7.3-54 package or the NBZIMM v.1.0 package depending on the model used. The significance threshold was likewise set at 0.05.

The R packages BiodiversityR version 2.14-1, PMCMRplus version 1.9.4, RVAideMemoire version 0.9-8, and vegan version 2.5-6 were used for various statistical analyses.

#### 5.1.6. Quantification of Uremic Toxins Using UPLC (Ultra-Performance Liquid Chromatography) in Plasma Samples of the Rat Experimental Model and of the Patients from the Clinical Study

Uremic toxins in plasma can be quantified as total and free fractions. The free fraction represents the unbound form of the toxin, which is not bound to albumin and is considered the biologically active fraction. In contrast, the total fraction includes both the protein-bound and unbound forms, reflecting the overall circulating concentration. For the total fraction, plasma samples (100 µL) were diluted with 260 µL of HPLC-grade water (Thermo Scientific, Waltham, MA, USA) and incubated for 30 min at 95 °C to denature proteins. Samples were then placed on ice for 10 min and centrifuged at 7379 *g* for 10 min. Subsequently, the supernatants were filtered using Amicon Ultra 0.5 mL filters (Merck KGaA, Darmstadt, Germany) at 3615 g for 20 min at room temperature (RT). Finally, 180 µL of the ultrafiltrate was transferred to an autosampler vial, and 20 µL of fluorescein (50 mg/L) was added as an internal standard.

To determine the free fraction of uremic toxins, plasma samples (260 µL) were first filtered using Amicon Ultra 0.5 mL filters (Merck KGaA, Darmstadt, Germany) at 3615 g for 20 min at room temperature. Then, 100 µL of the ultrafiltrate was diluted with 260 µL of HPLC-grade water (Thermo Scientific, Waltham, MA, USA) and incubated for 30 min at 95 °C for protein denaturation. Samples were placed on ice for 10 min and centrifuged at 7379 *g* for 10 min. Finally, 180 µL of the supernatant was transferred to an autosampler vial, and 20 µL of fluorescein (50 mg/L) was added as an internal standard.

All samples, including both total and free fractions, were stored at 4 °C, and 18 µL were injected into the column. An Agilent 1290 Infinity system (Agilent, Santa Clara, CA, USA) was used for quantifying uremic toxins. Chromatographic separation was performed at 26 °C using a Waters Acquity UPLC BEH C18 column (Milford, MA, USA) (1.7 µm, 100 × 2.1 mm) coupled with a Waters Acquity UPLC BEH C18 VanGuard pre-column (1.7 µm, 5 × 2.1 mm). The mobile phase consisted of 50 mM ammonium formate buffer, HPLC-grade water with formic acid (4.5 mL) and ammonia (1.4 mL) (mobile phase A, pH 3.0), and methanol (mobile phase B). HPLC-grade water and methanol were purchased from Acros Organics (Fair Lawn, NJ, USA). Formic acid and ammonia were obtained from VWR (Leuven, Belgium) and Merck (Merck KGaA, Darmstadt, Germany), respectively. A linear gradient elution was employed at a flow rate of 0.3 mL/min, starting with 98% A, transitioning to 90% A over 7 min. Over the next 9 min, the mobile phase shifted to 100% B and was maintained for 3 min, followed by a re-equilibration step.

Indoxyl sulfate (λex: 280 nm), p-cresyl sulfate and p-cresyl glucuronide (λex: 264 nm, λem: 290 nm), indole-3-acetic acid (λex: 280 nm, λem: 350 nm), and fluorescein as an internal standard (λex: 443 nm, λem: 512 nm) were detected using an Agilent G1316C fluorescence detector.

### 5.2. Clinical Study: Methods

In this randomized, open-label, parallel-group clinical trial, all subjects signed the informed consent form before inclusion in the study after the nature of the procedure had been fully explained to them. The name of the study was Mickid (Microbiota and kidney), the study was conducted under the Declaration of Helsinki, and the protocol was approved by the Ethics Committee of Cordoba (Cordoba Research Ethics Committee, Spain. Record number: 4922, Committee file number: 332), with ClinicalTrials.gov Identifier NCT07260682.

The sample size was calculated using the Grammo software, with the primary variable being changes in the degree of VC as assessed by the Kauppila and Adragao scoring methods. A subject was classified as ‘withdrawn’ from the study if they did not follow the study procedure, had not undergone follow-up, or if no further information was available since the withdrawal date or the last contact. Twenty-three patients with advanced CKD (stage 5) and VC were recruited and randomized to either receive or not receive the same probiotic used in the experimental study, adjusted to a dose of 15 mL/day orally, containing 10^9^ CFU/mL, for 6 months. Six of the patients dropped out of the study (Appendix A). So, in our exploratory clinical trial, all participants were at stage 5 of CKD. Within this group, some patients were undergoing dialysis treatment. Specifically, in the VC group (n = 11), three patients were receiving peritoneal dialysis (PD) and two were undergoing hemodialysis (HD). In the VC + Prob group (n = 6), one patient was under HD treatment.

In this clinical study, treatment allocation followed an open-label design, but radiological assessments of VC (X-ray and CT) were carried out by radiologists blinded to group assignment.

Inclusion and exclusion criteria of this study are shown in Appendix A. All patients were followed in the outpatient clinic on the nephrology service of the Reina Sofia University Hospital in Córdoba (Spain). Baseline data included age, gender, body mass index, blood pressure, and comorbidities or prevalent diseases. Baseline comorbid conditions were characterized based on the KDIGO and global clinical practice guidelines. Hypertension (HT) was identified as systolic blood pressure ≥ 130 mmHg and/or diastolic blood pressure ≥ 80 mmHg, or current use of antihypertensive medications. Diabetes mellitus (DM) was recognized as a clinical diagnosis, characterized by plasma glucose levels ≥ 126 mg/dL, HbA1c levels ≥ 6.5%, or the administration of antidiabetic drugs. A diagnosis or treatment with lipid-lowering agents in line with KDIGO guidelines for lipid management determined Dyslipidemia (DL). Hyperuricemia (HU) was defined as serum uric acid levels > 7.0 mg/dL in men and >6.0 mg/dL in women, or current hypouricemic treatment. Smoking status was classified as a current smoker. Ischemic heart disease (IHD) and cardiovascular disease (CVD) were defined based on documented medical history. Dialysis status and modality (hemodialysis (HD) or peritoneal dialysis (PD)) were recorded at baseline. All patients had normal and stable values of blood pressure, glycemia, and P before entering the study, and no treatment adjustment was required throughout the experimental period. Urine, blood, and stool samples were collected at baseline visit (V1) and final (V3), corresponding to months 0 and 6 of the study. To monitor compliance, probiotic containers were provided to the patients, who were asked to return them at each follow-up visit. The remaining quantity was checked to verify appropriate consumption (Figure 12).

#### 5.2.1. Estimation of Nutritional Status

Phosphate, calcium, and protein intake, as well as the amount of processed food and soft drinks, were estimated using a table that includes the composition of the foods consumed in the south of Spain. The information on the content of P shown in this table coincides with the values indicated in other sources of food information, the Spanish Database of Food Composition “BEDCA” [Spanish Food Composition Database published by the BEDCA Network of the Ministry of Science and Innovation. Available online: https://www.bedca.net/bdpub/ (accessed on 12 July 2020)]. The food composition values collected in this database have been obtained from different sources, including laboratories, the food industry, and scientific publications. This database was built according to the European standards developed by the EuroFIR European Network of Excellence and is included in the list of food composition databases of the EuroFIR Association. The dietary records are considered an appropriate method for assessment of dietary intake by the European Food Safety Authority (EFSA) pan-European dietary survey. However, the data collected is an estimate based on the self-reported number of processed products eaten by each patient through the dietary survey. The method used to assess food intake is based on two tools: the Diet Calibrator and the Spanish Food Composition Database (BEDCA). Both methods have already been validated in previous studies by our research group, as described by Pendón-Ruiz de Mier et al. and Novillo et al. in Nutrients (2021 and 2025, respectively) [74,75].

#### 5.2.2. Blood and Urine Analysis

Blood was collected at baseline and after 6 months for measurement of standard plasma biochemistry, the complete blood count, and mineral metabolism parameters. The eGFR was calculated by the CKD-EPI formula [76]. ELISA assay was used for the measurement of plasma iFGF23. Urine samples were collected for quantification of protein and creatinine using an Architect c-16000 device (Abbott^®^, Chicago, IL, USA).

#### 5.2.3. Assessment of Vascular Calcification in Patients

X-rays of the pelvis and both hands were performed to assess and score the presence of VC according to Adragao index, and an X-ray of the lumbar spine, to Kauppila index. It was also carried out a non-contrast CT with quantification of mass and volume of calcium in L1–L5.

#### 5.2.4. Inflammatory Profile in Patients

Inflammatory profile was evaluated in serum samples by performing a proximity extension assay (PEA), subjected to high-throughput analysis of 92 inflammation-associated proteins. The panel used was Olink Target 96 Inflammation panel from Cobiomic Bioscience S.L. (Córdoba, Spain). All the inflammatory parameters measured are shown in Appendix A.

### 5.3. Statistical Analysis

Continuous variables are shown as mean ± standard deviation (SD), and the content of calcium in rats’ thoracic aorta is shown as mean ± standard error of the mean (SEM). Patients are shown as mean ± SD or median (interquartile range, IQR). Categorical variables are presented as a percent (%). Statistical differences between the groups were assessed by *t*-test or the corresponding non-parametric test, after performing the normality test. A *p*-value < 0.05 was considered statistically significant. Statistical analysis was performed using the SPSS statistical program (IBM SPSS Statistics 25), graphs with GraphPad Prism 8.0.2, MetaboAnalyst 6.0, correlogram with R 4.4.1, RStudio, and LDA with R 4.4.0 and R Studio.

## Figures and Tables

**Figure 1 toxins-18-00006-f001:**
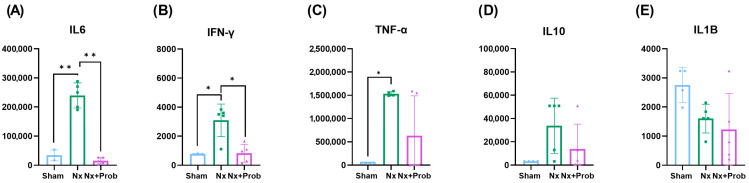
Inflammation in the rat experimental model. Five inflammatory cytokines were studied: (**A**) interleukin-6 (IL-6), (**B**) interpheron-γ (IFN-γ), (**C**) tumor necrosis factor-α (TNF-α), (**D**) interleukin-10 (IL-10), and (**E**) interleukin-1β (IL-1β). Statistical significance is indicated as * *p* ˂ 0.05 and ** *p* ˂ 0.01. Variables are shown as mean ± standard deviation (SD) of the non-parametric test.

**Figure 2 toxins-18-00006-f002:**
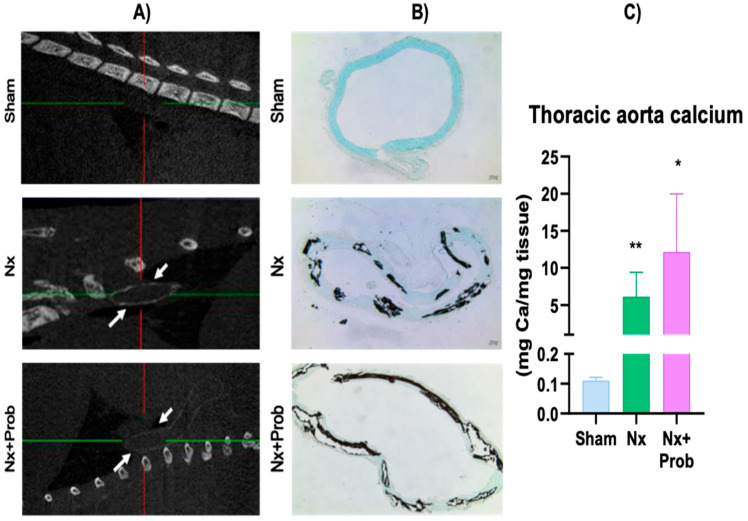
Vascular calcification in the rat experimental model. Representative computed tomography (micro-CT) images of sagittal sections of the thorax of the studied groups (**A**). At the intersection of the colored lines, hyperdense areas can be observed, which correspond to VC (white arrows). Representative images of Von Kossa histological staining in the thoracic aorta of the different experimental groups (**B**). Mineral content of calcium in the thoracic aorta (**C**). Data are shown as mean ± standard error of the mean (SEM) of the non-parametric test. Sham n = 9, Nx n = 14, Nx + Prob n = 5. Sham * *p* ˂ 0.05 and ** *p* ˂ 0.01 vs. Sham. Nx: 5/6 nephroctomy; Prob: probiotic.

**Figure 3 toxins-18-00006-f003:**
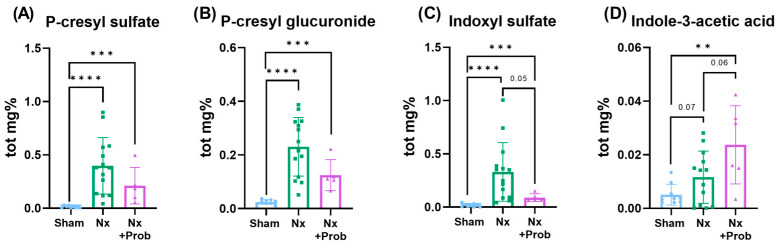
Plasma levels of microbiota-derived uremic toxins in experimental groups. Bar graphs represent total concentrations of p-cresyl sulfate (**A**), p-cresyl glucuronide (**B**), indoxyl sulfate (**C**), and indole-3-acetic acid (**D**) in plasma from Sham (control), Nx (5/6 nephrectomized), and Nx + Prob (5/6 nephrectomized treated with probiotics) rats. Data are shown as mean ± standard deviation (SD) of the non-parametric test, with individual data points indicated. Statistical significance is indicated as ** *p* < 0.01, and *** *p* < 0.001 and **** *p* < 0.0001.

**Figure 4 toxins-18-00006-f004:**
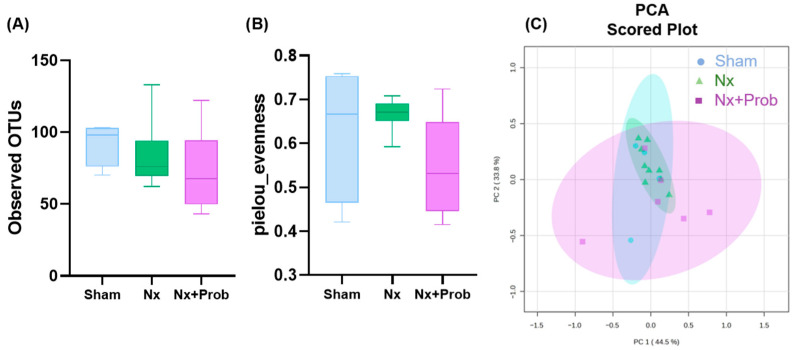
Bacterial diversity in the rat experimental model. Alpha diversity. Within each sample, the richness, represented by the number of distinct OTUs (**A**), and the evenness of abundance measured by Pielou’s index are shown (**B**). Beta diversity was analyzed by Unsupervised Principal Component Analysis, PCA (**C**). Each point in the graph represents a sample, and the points with the same color are from the same group. The distance between any pair of points indicates the difference in gut microbial composition between the two samples. Sham n = 4, Nx n = 9, Nx + Prob n = 6. Nx: 5/6 nephrectomy; Prob: probiotic. No significant differences were observed between groups.

**Figure 5 toxins-18-00006-f005:**
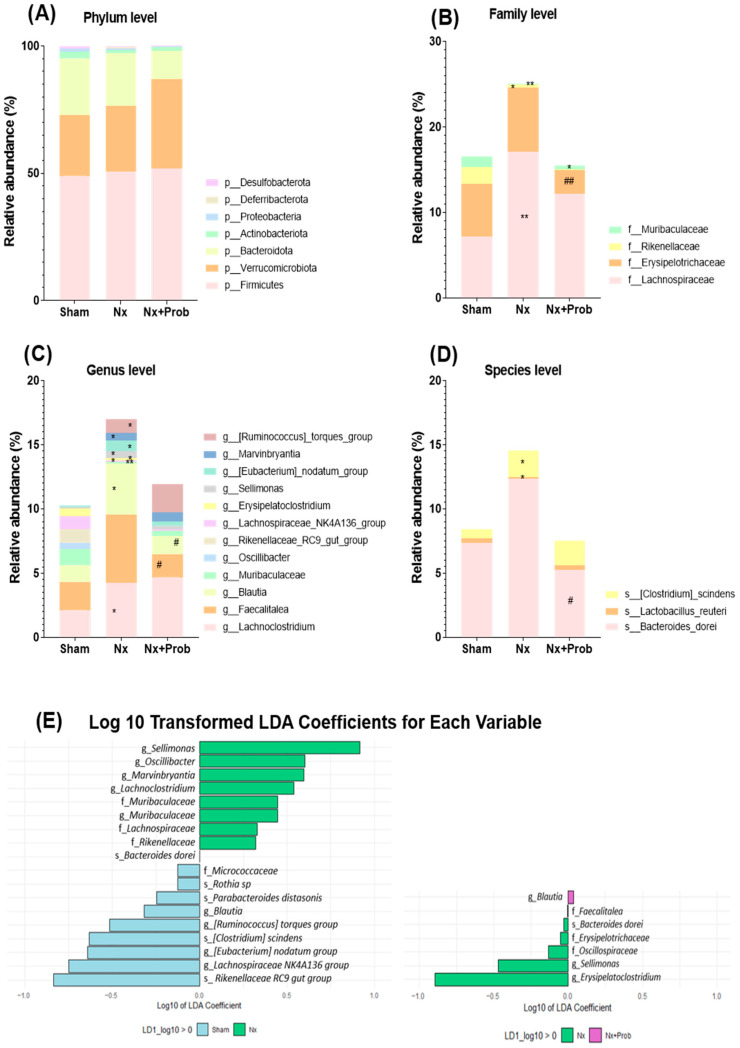
Taxonomic profile and LDA in the rat experimental model. Relative abundance by group at phylum level (**A**), family (**B**), genus (**C**) and species (**D**). Panels (**B**–**D**) only present the families, genera and species for which a significantly different relative abundance was found. Statistical significance is indicated as * *p* < 0.05 and ** *p* < 0.01 vs. Sham, # *p* < 0.05 and ## *p* < 0.01 vs. Nx. Log10 transformed LDA coefficients for each variable, highlighting significant differences in bacterial genera between Sham (n = 4), Nx (n = 9), and Nx + Prob (n = 6) groups (**E**). Nx: 5/6 nephrectomy; Prob: probiotic; p: phylum; f: family; g: genus; s: species.

**Figure 6 toxins-18-00006-f006:**
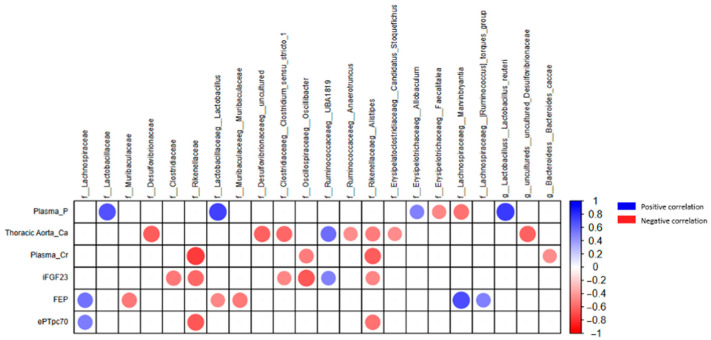
Correlogram between kidney function and bacterial taxa parameters in the rat experimental model. Spearman correlation matrix showing significant relationships between different bacterial taxa and clinical parameters in the study groups. Plasma_P (plasma phosphate), Thoracic Aorta Ca (calcium at the thoracic aorta), Plasma_Cr (plasma creatinine), iFGF23 (intact fibroblast growth factor 23), FEP (fractional excretion of phosphate), and ePTpc70 (end-proximal tubule phosphate concentration). Red dots represent negative correlations, and blue dots represent positive correlations. As the color intensity increases and the dot size grows, the strength of the correlation becomes greater. Sham n = 4, Nx n = 9, Nx + Prob n = 6.

**Figure 7 toxins-18-00006-f007:**
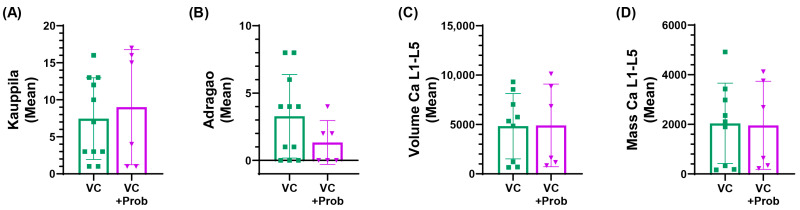
Assessment of vascular calcification after 6 months of follow-up in patients with CKD and VC enrolled in the exploratory clinical study. Vascular calcifications were quantified using plain radiography with Kauppila (**A**) and Adragao (**B**) scores. Calcium volume (**C**) and mass (**D**) between the L1-L5 vertebrae were quantified using non-contrast CT. No significant differences were observed between groups.

**Figure 8 toxins-18-00006-f008:**
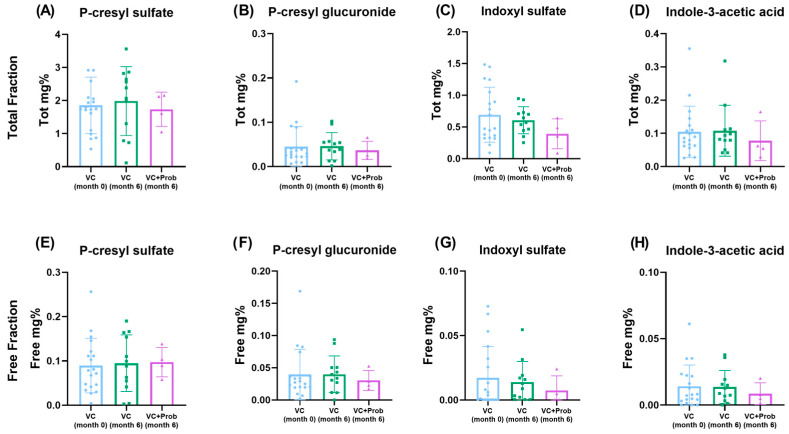
Plasma levels of microbiota-derived uremic toxins after 6 months of follow-up in patients with CKD and VC enrolled in the exploratory clinical study. Bar graphs show total (top row **A**–**D**) and free (bottom row **E**–**H**) fraction of the following uremic toxins, respectively: P-cresyl sulfate, P-cresyl glucuronide, indoxyl sulfate, and indole-3-acetic acid in control (VC, month 0), vascular calcification (VC, month 6), and vascular calcification with probiotic treatment (VC + Prob, month 6) groups. Data are presented as mean ± SD; individual data points are shown. No significant differences were observed between groups for any of the measured metabolites.

**Figure 9 toxins-18-00006-f009:**
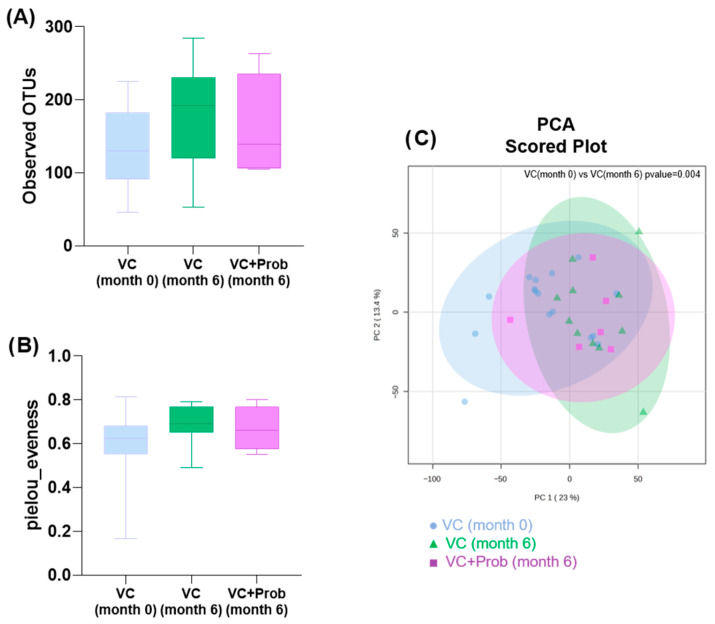
Bacterial diversity after 6 months of follow-up in patients with CKD and VC enrolled in the exploratory clinical study. Alpha diversity showing richness (**A**) and evenness (**B**). Beta diversity analyzed through PCA (**C**), with significant differences between VC group at month 0 and month 6 (*p* = 0.004).

**Figure 10 toxins-18-00006-f010:**
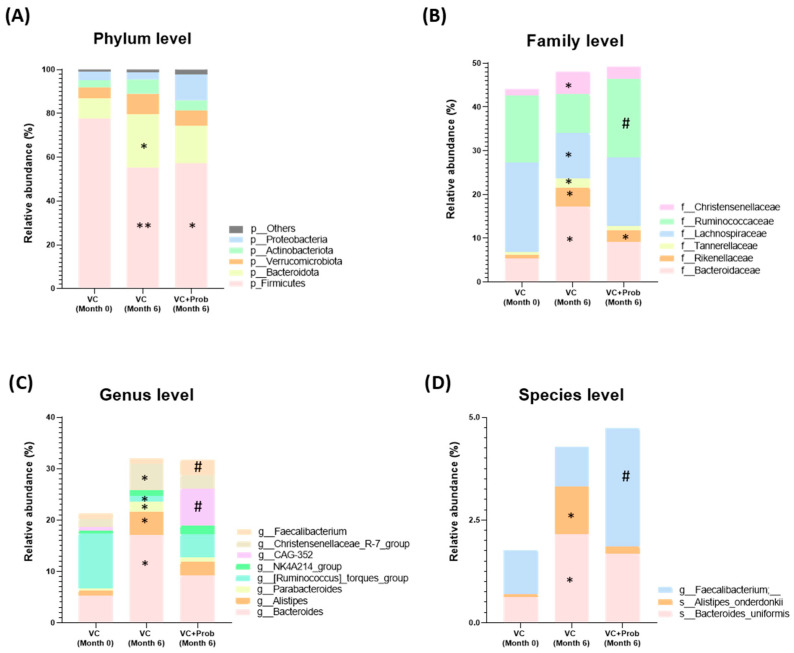
Changes in taxonomic profile after 6 months of follow-up in calcified CKD patients enrolled in the exploratory clinical study. Longitudinal analysis of bacterial taxonomy in advanced CKD and CV patients with and without probiotic administration. Panels (**B**–**D**) only present the families, genera, and species for which a significantly different relative abundance was found. Comparisons are shown at the (**A**) phylum, (**B**) family, (**C**) genus, and (**D**) species levels. Statistical significance is indicated as * *p* < 0.05 and ** *p* < 0.01 vs. CV (month 0) # *p* < 0.05 vs. CV (month 6).

**Figure 11 toxins-18-00006-f011:**
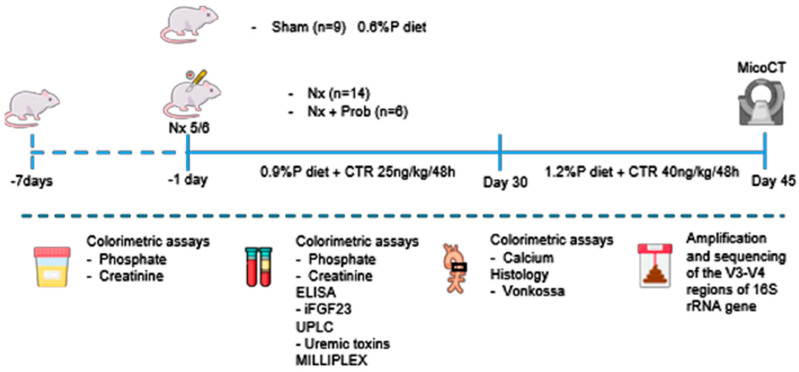
In vivo study design. Schematic representation of the experimental timeline and interventions in the study groups. Uremia was induced through a 5/6 nephrectomy (Nx). Phosphate content of the diet was 0.9% and the dose of calcitriol was 25 ng/kg/48 h for 30 days. Subsequently, for the following 15 days, the dietary phosphate content was increased to 1.2%, accompanied by calcitriol administration at 40 ng/kg/48 h. Rats were divided into three groups: Sham (saline by oral gavage, n = 9), Nx (saline by oral gavage, n = 14) and Nx + Prob (probiotic through an oral gavage, n = 6).

**Figure 12 toxins-18-00006-f012:**
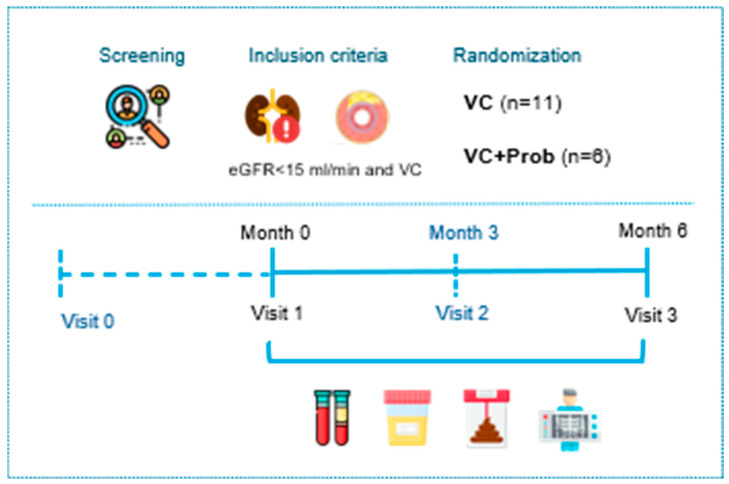
Study design of the clinical trial. Overview of the clinical trial design, including screening (visit 0), randomization (visit 1), follow-up period (visit 2), and the final visit (visit 3), with assessments of renal function, inflammation, microbiota characterization, and vascular calcification. Urine, blood, and stool samples were collected at visits 1 and 3.

**Table 1 toxins-18-00006-t001:** Biochemical characteristics of the different experimental groups.

	Sham	Nx	Nx + Prob
*n*	9	14	6
*Plasma P (mg/dL)*	3.90 ± 0.859	7.10 ± 3.367 *	5.97 ± 4.238
*Plasma Cr (mg/dL)*	0.80 ± 0.141	1.56 ± 0.475 ***	1.93 ± 1.208 **
*Urine Cr (mg/24 h)*	11.01 ± 3.606	9.26 ± 2.438	7.23 ± 2.141 *
*FEP (%)*	53.0 ± 31.99	186.8 ± 89.65 ***	169.1 ± 133.8 *
*ePTpc70 (mg/dL)*	1.84 ± 0.735	11.62 ± 4.565 ***	7.80 ± 4.196 *
*iFGF23 (pg/mL)*	124.9 ± 51.92	22,586 ± 31,304 **	30,468 ± 28,336 **

Plasma and urine parameters measured in Sham, 5/6 nephrectomy (Nx), and Nx + Probiotic (Nx + Prob) groups, including plasma phosphate (P), plasma creatinine (Cr), urinary creatinine, fractional excretion of phosphate (FEP), end-proximal tubule phosphate concentration (ePTpc70) and intact fibroblast growth factor 23 (iFGF23). Statistical significance is indicated as * *p* < 0.05, ** *p* < 0.01, and *** *p* < 0.001 vs. Sham group. Variables are shown as mean ± standard deviation (SD) of the non-parametric test.

**Table 2 toxins-18-00006-t002:** Morphometric parameters and comorbidities.

	Control	VC	VC + Prob	*p*-Value
** *n* **	10	11	6	-
** *Age (years)* **	61 ± 5	67 ± 10	64 ± 14	0.314
** *BMI (kg/m^2^)* **	26.7 ± 4.35	28.0 ± 7.07	27.4 ± 3.84	0.872
** *Systolic BP (mmHg)* **	117.1 ± 16.62	150.3 ± 25.02 a	143.5 ± 26.20 a	**0.007**
** *Diastolic BP (mmHg)* **	78.2 ± 7.50	81.6 ± 15.55	79.3 ± 13.00	0.817
** *PP (mmHg)* **	38.7 ± 11.19	68.5 ± 23.61 a	64.0 ± 27.12	**0.008**
** *Male (n/%)* **	6/60	9/82	3/50.0	0.352
** *HT (n/%)* **	0/0	10/91 a	6/100 a	**0.000**
** *DM (n/%)* **	0/0	6/55 a	3/50 a	**0.019**
** *DL (n/%)* **	2/20	9/82 a	6/100 a	**0.001**
** *HU (n/%)* **	0/0	10/91 a	5/83 a	**0.000**
** *Smoke (n/%)* **	0/0	2/18	1/17	0.101
** *IHD (n/%)* **	0/0	3/27	1/17	0.211
** *CVD (n/%)* **	0/0	2/18	0/0	0.208
** *Dialysis (n/%)* **	0/0	5/46 (PD:3; HD:2)	1/17 (HD:1)	0.109

Baseline characteristics of the CKD patients with VC either with (VC + Prob) or without (VC) probiotic administration. Including age, body mass index (BMI), Systolic (SBP) and diastolic (DBP) blood pressure, pulse pressure (PP) and prevalence of comorbidities such as hypertension (HT), diabetes (DM), dyslipidemia (DL), hyperuricemia (HU), smoking status, ischemic heart disease (IHD), cardiovascular disease (CVD) and dialysis (peritoneal dialysis: PD; hemodialysis: HD). a *p* < 0.05 vs. Control group.

**Table 3 toxins-18-00006-t003:** Changes in plasma parameters of kidney function and mineral metabolism in patients with CKD and VC, either with (VC + Prob) or without (VC) probiotic administration over the study period of 6 months.

	VC (M 0)	VC (M 6)	VC + Prob (M 0)	VC + Prob (M 6)	*p*-Value
** *Cr (mg/dL)* **	4.02 ± 1.56	4.50 ± 1.06	3.85 ± 0.77	5.17 ± 2.28	0.398
** *GFR (mL/min)* **	18.67 ± 4.5	16.50 ± 3.72	17.20 ± 3.35	16.67 ± 3.20	0.711
** *Mg (mg/dL)* **	2.11 ± 0.37	2.05 ± 0.29	2.17 ± 0.15	2.25 ± 0.36	0.637
** *Ca (mg/dL)* **	9.36 ± 0.86	9.06 ± 0.47	9.12 ± 0.63	8.60 ± 0.64	0.202
** *P (mg/dL)* **	5.02 ± 1.50	4.26 ± 1.05	4.48 ± 1.00	5.08 ± 1.34	0.474
** *1,25OHD (pg/mL)* **	23.36 ± 14.49	21.50 ± 10.14	16.75 ± 4.86	18.67 ± 9.93	0.744
** *25OHD (ng/mL)* **	26.78 ± 17.09	28.16 ± 9.66	30.07 ± 20.27	23.85 ± 4.52	0.893
** *PTH (pg/mL)* **	237.24 ± 146.66	245.05 ± 133.59	348.78 ± 304.03	427.25 ± 314.75	0.315
** *iFGF23 (pg/mL)* **	1242.29 ± 1329.72	1133.99 ± 1154.37	405.21 ± 328.29	843.16 ± 886.76	0.464
** *Prot/Cr (mg/g)* **	1.80 ± 0.57	1.55 ± 0.53	1.84 ± 0.47	1.59 ± 0.78	0.782
** *Hb (g/dL)* **	12.20 ± 1.02	11.86 ± 1.55	11.73 ± 0.93	11.67 ± 1.98	0.855
** *Alb (g/dL)* **	4.34 ± 0.34	4.03 ± 0.39	4.32 ± 0.30	4.15 ± 0.36	0.232
** *CRP (mg/L)* **	2.27 ± 3.13	15.5 ± 33.73	4.00 ± 7.83	5.86 ± 5.57	0.461
** *Iron (mg/dL)* **	88.55 ± 46.14	56.13 ± 18.36	81.75 ± 28.88	68.00 ± 38.26	0.418
** *Ferritin (ng/mL)* **	185.55 ± 196.90	213.162 ± 247.07	230.90 ± 355.09	54.04 ± 47.83	0.592

Includes data on creatinine (Cr), glomerular filtration rate (GFR), magnesium (Mg), calcium (Ca), phosphate (P), vitamin D (1,25OHD, 25OHD), parathyroid hormone (PTH), intact Fibroblast Growth Factor-23 (iFGF23), protein/creatinine ratio, hemoglobin (Hb), albumin (Alb), c-reactive protein (CRP), iron, and ferritin levels. M 0 is month 0 and M 6 corresponding with month 6.

**Table 4 toxins-18-00006-t004:** Changes in nutritional status in the exploratory clinical study measured after dietary survey.

*DIET*	VC (M 0)	VC (M 6)	VC + Prob (M 0)	VC + Prob (M 6)	*p*-Value
** *Phosphate* **	1058 ± 286.9	976.6 ± 250.8	982.1 ± 346.7	997.9 ± 144.3	0.943
** *Calcium* **	688.5 ± 213.1	552.6 ± 174.6	650.4 ± 305.8	738 ± 126.3	0.382
** *Protein* **	69,683 ± 20,270	62,099 ± 17,705	60,075 ± 11,152	59,453 ± 17,237	0.698
** *Processed food* **	64,185 ± 53,359	74,999 ± 78,017	82,138 ± 91,938	91,533 ± 92,831	0.911
** *Soft drinks* **	24.44 ± 73.33	0 ± 0	55 ± 134.7	33 ± 73.79	0.581

Changes in dietary patterns over time or between groups regarding phosphate, calcium, protein, processed foods, or soft drinks consumed in the diet. M 0 is month 0 and M 6 corresponds with month 6.

## Data Availability

The datasets generated during the current study are available in the European Nucleotide Archive (ENA) repository, Project accession: PRJEB87598.

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
