# Peer review of "Probiotic Supplementation in Chronic Kidney Disease: Outcomes on Uremic Toxins, Inflammation, and Vascular Calcification from Experimental and Clinical Models"

_toxins, 2025, doi:10.3390/toxins18010006_

Round 1
Reviewer 1 Report (Previous Reviewer 1)
Comments and Suggestions for Authors
The present study is very interesting because it combines experimental and clinical studies. From a scientific point of view, it brings up a very relevant topic about the effects of probiotic supplementation from different aspects. I would like to congratulate the authors
In the abstract the acronym IS appears for the first time, not being cited
To what do the authors attribute the non-reduction of inflammatory parameters in the clinical study?
Did the clinical study include patients undergoing conservative treatment and hemodialysis? That wasn't very clear.
But if there are patients at different stages, does this influence any parameters? It would no longer make sense to separate them by stage of the disease?
What are the expected results?
The number of participants was very small, but the 6 months of supplementation was interesting. How did the authors control the consumption of probiotics by patients?
The method for assessing food intake was not the best, what was the reason for choosing it?
Author Response
Thank you very much for taking the time to review this manuscript. Please find the detailed responses below and the corresponding corrections highlighted in red in the new version of the manuscript.
Reviewer 1
The present study is very interesting because it combines experimental and clinical studies. From a scientific point of view, it brings up a very relevant topic about the effects of probiotic supplementation from different aspects. I would like to congratulate the authors.
We thank Reviewer 1 for their kind words and for their consideration of our work.
Comments 1: In the abstract the acronym IS appears for the first time, not being cited
Response 1: Thank you very much for your appreciation. We have now added the full term followed by its abbreviation (IS) at its first mention in the abstract.
Comments 2: To what do the authors attribute the non-reduction of inflammatory parameters in the clinical study?
Response 2: Thank you for this interesting question. Given that our patients are in a very advanced stage of chronic kidney disease and with vascular calcification, they have a persistently elevated status of chronic inflammation. We believe that, under these conditions, the anti-inflammatory stimulus provided by the probiotic may not be sufficient to counteract or substantially reduce the pre-existing chronic inflammation, at least with this formulation, this specific probiotic, and this study design.
This point has been incorporated into the Discussion section of the manuscript, page 14, paragraph 5, and lines 580-583.
“The high pro-inflammatory state characteristic of patients with CKD may explain the limited effect of the probiotic on the inflammatory status. The administration of a probiotic like this might be insufficient to elicit measurable beneficial effects within the time frame and at the dose evaluated in this study”.
Comments 3: Did the clinical study include patients undergoing conservative treatment and hemodialysis? That wasn't very clear.
Response 3: We appreciate this observation and have clarified this point in the revised manuscript. In our exploratory clinical trial, all participants were at stage 5 of chronic kidney disease (CKD). Within this group, some patients were undergoing dialysis treatment. Specifically, in the “VC” group (n = 11), three patients were receiving peritoneal dialysis and two were undergoing hemodialysis. In the “VC+Prob” group (n = 6), one patient was under hemodialysis treatment.
The following information has been included in Table 2 (page 8, line 319) and in the Methods section (page 21, paragraph 2, and lines 880-884):
“So, in our exploratory clinical trial, all participants were at stage 5 of CKD. Within this group, some patients were undergoing dialysis treatment. Specifically, in the VC group (n = 11), three patients were receiving peritoneal dialysis (PD) and two were undergoing hemodialysis (HD). In the VC+Prob group (n = 6), one patient was under HD treatment.”
Comments 4: But if there are patients at different stages, does this influence any parameters? It would no longer make sense to separate them by stage of the disease?
Response 4: Thank you for this valuable comment. As mentioned above, all our participants were classified as stage 5 CKD patients; therefore, stratification by disease stage was not applicable in this cohort, and the results can be considered homogeneous in terms of disease severity. Moreover, due to the small number of patients in our study, stratification, such as comparing different types of dialysis, significantly reduces the statistical power because of the limited sample size within each group. In contrast, the paired analysis including all patients provides greater statistical power, allowing us to draw more rigorous conclusions about the effect of treatment with our probiotic over 6 months under the conditions described.
Comments 5: What are the expected results?
Response 5: Our initial hypothesis was that probiotic administration could modulate the gut microbiota, thereby reducing systemic inflammation and uremic toxin levels, ultimately leading to decreased vascular calcification. However, our findings indicate that probiotic supplementation did not reduce vascular calcification in either the experimental model or the exploratory clinical trial. This lack of effect may be due to the severity of tissue damage and inflammation in both settings, which may be too substantial to be mitigated by a single probiotic administered for six months.
Comments 6: The number of participants was very small, but the 6 months of supplementation was interesting. How did the authors control the consumption of probiotics by patients?
Response 6: We appreciate this relevant question. To monitor compliance, probiotic containers were provided to the patients, who were asked to return them at each follow-up visit. The remaining quantity was checked to verify appropriate consumption. Furthermore, microbiota analysis revealed an increased relative abundance of the Ruminococcaceae family, present in the probiotic formulation, at the family, genus, and species levels, supporting adherence to the supplementation protocol and serving as a biological tracer. In the animal study, to ensure full compliance, probiotics were administered via oral gavage.
The following information has been included in the Methods section (page 21, paragraph 4, and lines 895-897):
“To monitor compliance, probiotic containers were provided to the patients, who were asked to return them at each follow-up visit. The remaining quantity was checked to verify appropriate consumption.”
Comments 7: The method for assessing food intake was not the best, what was the reason for choosing it?
Response 7: We understand the reviewer's concern. The method used to assess food intake was chosen because it has been previously validated and used in similar studies in our research group, as described in Nutrients (2021), doi: 10.3390/nu13020292 and Nutrients (2025) doi:10.3390/nu17213323. This method estimated food intake using two tools: the Diet Calibrator and the Spanish Food Composition Database (BEDCA).
This information has been included in the methods section of the new versión of the manuscript (Page 22, paragraph 1, and lines 923-927):
“The method used to assess food intake are based on two tools: the Diet Calibrator and the Spanish Food Composition Database (BEDCA). Both methods, have already been validated in previous studies by our research group, as described in Nutrients (2021), doi: 10.3390/nu13020292 and Nutrients (2025) doi:10.3390/nu17213323.”

Reviewer 2 Report (New Reviewer)
Comments and Suggestions for Authors
In this work, the authors revealed the effects of probiotics on kidney function, uremic toxins, and vascular calcification in CKD based on experimental and human studies. Several suggestions are made as follows to improve the quality of the manuscript.
- The title makes no sense. Please change the title
- The abstract should be improved. Please more result in details in abstract.
- All abbreviations should be substantiated for the first time in the abstract.
- All abbreviations should be substantiated for the first time in the main text.
- Please provide the references for “CKD patients experience elevated oxidative stress, chronic low-grade inflammation, and accumulation of uremic toxins.”
- The authors always put many references together, such as [2-8], [9-14], [15-17]. However, there is lack of many references in each paragraph, such as “It is well established that in patients with CKD…, thus pushing through VC.”
- The authors investigated the impact of a probiotic supplementation on kidney function, mineral metabolism, inflammation and VC in both an experimental rat model and patients with advanced CKD and VC.
- The description “In recent years, the study of the microbiota and its potential therapeutic applications in various diseases has gained considerable interest [15-17]” is irrelevant to this research. The reviewer suggests removing this sentence. The authors should introduce the CKD and the dysbiosis of gut microbiota base on publications, such as Bibliometric analysis of the relationship between gut microbiota and chronic kidney disease from 2001-2022, PMID: 39663206, PMID: 40761666.
- The authors demonstrated that probiotic supplementation decreased inflammation and tented to decrease IS levels in the experimental model. However, did not lead to significant improvements in kidney function and VC neither rats nor patients. The latest several studies reported Targeting Lactobacillus johnsonii reverse chronic kidney disease, Probiotic interventions in peritoneal dialysis: A review of underlying mechanisms and therapeutic potentials.
- Please provide both manufacturer’s name and location (city, state, and country) for important equipment and reagents in the manuscript.
- The total concentrations of pCS, pCG, IS, and IAA showed statistically significant differences across different groups, but in Figure 4, bacterial diversity in the rat experimental model did not show a statistically significant difference. How can this result be explained?
- Previous studies show that these metabolites are associated with renal functions in serum or renal tissue of UUO and CKD patients, such as “A possible role of p-cresyl sulfate and indoxyl sulfate as biomarkers in the prediction of renal function according to the GFR (G) categories”, “Tryptophan metabolism as a target in gut microbiota, ageing and kidney disease”. The authors demonstrated that probiotic supplementation decreased the plasma concentrations of the total fraction of pCS, pCG, IS, and IAA in NX rats, while probiotic supplementation decreased the total concentrations of pCS, pCG, IS, and IAA. These results are paradoxical with previous publications. The reviewer suggests discussing and comparing this study with previous publications.
- The authors demonstrated that probiotic supplementation decreased the plasma concentrations of the total fraction of pCS, pCG, IS, and IAA in NX rats, while probiotic supplementation decreased the total concentrations of pCS, pCG, IS, and IAA. The underlying molecular mechanism of this result is unclear. Reviewers suggested that the authors elucidate the potential molecular mechanism through in vitro and in vivo experiments. Alternatively, authors could discuss and speculate on potential molecular mechanisms based on previously publications “Intrarenal 1-methoxypyrene, an aryl hydrocarbon receptor agonist, mediates progressive tubulointerstitial fibrosis in mice”, “Lactobacillus species ameliorate membranous nephropathy through inhibiting aryl hydrocarbon receptor pathway via tryptophan‐produced indole metabolites”. For example, pCS, pCG, IS, and IAA are all endogenous ligands of aryl hydrocarbon receptors, and some studies have demonstrated that these metabolites, acting as ligands for aryl hydrocarbon receptors, activate aryl hydrocarbon receptor signaling, mediating kidney injury and renal fibrosis.
- The clinical studies should be discussed and improved.
- The latest publications should be further discussed.
- Please change the references based on the guide for authors.
- The references 2-4, 37, 45, 71 are out of date. Please remove or cite the latest publications.
Author Response
Thank you very much for taking the time to review this manuscript. Please find the detailed responses below and the corresponding corrections highlighted in red in the new version of the manuscript.
Reviewer 2
In this work, the authors revealed the effects of probiotics on kidney function, uremic toxins, and vascular calcification in CKD based on experimental and human studies. Several suggestions are made as follows to improve the quality of the manuscript.
Comments 1: The title makes no sense. Please change the title
Response 1: We appreciate the suggestion. The title has been modified to “Probiotic Supplementation in Chronic Kidney Disease: Outcomes on Uremic Toxins, Inflammation, and Vascular Calcification from Experimental and Clinical Models.” (Page 1, paragraph 1, and lines 1-4.)
Comments 2: The abstract should be improved. Please more result in details in abstract.
Response 2: Thank you for your suggestion. We have revised the abstract to include more specific details about the main findings, including quantitative results and key outcomes from both the experimental and clinical components of the study. Page 1, paragraph 4, and lines 25-41.
Abstract:
Chronic kidney disease (CKD) is associated with gut microbiota alterations that contribute to increased inflammation, generation of uremic toxins, and may worsen the disease progression. While probiotics may improve the pro-inflammatory cytokine profile, their effects on mineral metabolism, vascular calcification (VC), and CKD progression remain unclear. We aimed to evaluate the impact of a commercial probiotic (Probimel) supplementation on kidney function, mineral metabolism, inflammation and VC in both an experimental rat model and patients with advanced CKD and VC. The experimental model of VC was performed through 5/6 nephrectomy (Nx), a high-phosphate diet, and calcitriol. Dividing animals into three groups: Sham, Nephrectomy, and Nephrectomy+Probiotic. In the exploratory clinical study, 23 patients with advanced stage 5 CKD and VC were randomized to receive or not daily probiotics for 6 months. Kidney function, mineral metabolism, uremic toxins, inflammation, VC, and fecal microbiota were evaluated. Probiotic supplementation decreased interleukin-6, (IL-6) and interpheron-γ (IFN-γ) and the uremic toxin, indoxyl sulfate (IS), levels in the experimental model. However, this probiotic did not improve kidney function and VC neither rats nor patients. Under our experimental and clinical conditions, the selected probiotic did not modify key parameters related to CKD progression or VC.
Comments 3: All abbreviations should be substantiated for the first time in the abstract.
Response 3: We appreciate this observation. We have carefully reviewed all abbreviations appearing in the abstract and have added the full term the first time each abbreviation is mentioned.
Comments 4: All abbreviations should be substantiated for the first time in the main text.
Response 4: Thank you for pointing this out. We have reviewed all abbreviations throughout the main text and have ensured that each term is written in full the first time it appears.
Comments 5: Please provide the references for “CKD patients experience elevated oxidative stress, chronic low-grade inflammation, and accumulation of uremic toxins.”
Response 5: We thank the reviewer for this comment. We have included the following references to support this claim (Frak W. 2024 doi: 10.3390/antiox13060687 and Harlacher E. 2022, doi: 10.3390/ijms23010531).
Comments 6: The authors always put many references together, such as [2-8], [9-14], [15-17]. However, there is lack of many references in each paragraph, such as “It is well established that in patients with CKD…, thus pushing through VC.”
Response 6: We understand and appreciate the reviewer's concern. We have reduced significantly the number of references and have revised the distribution of references throughout the manuscript to ensure that citations are placed at the end of each corresponding statement, rather than grouped only at the end of paragraphs. This provides clearer support for specific statements and improves readability.
Comments 7: The authors investigated the impact of a probiotic supplementation on kidney function, mineral metabolism, inflammation and VC in both an experimental rat model and patients with advanced CKD and VC.
Response 7: Exactly, that is the primary objective of our study, which seeks to investigate, in both preclinical and clinical settings, the translational potential of probiotic supplementation.
Comments 8: The description “In recent years, the study of the microbiota and its potential therapeutic applications in various diseases has gained considerable interest [15-17]” is irrelevant to this research. The reviewer suggests removing this sentence. The authors should introduce the CKD and the dysbiosis of gut microbiota base on publications, such as Bibliometric analysis of the relationship between gut microbiota and chronic kidney disease from 2001-2022, PMID: 39663206, PMID: 40761666.
Response 8: We agree with the reviewer. The suggested sentence has been removed and we have added a new text introducing the concept of gut microbiota dysbiosis and its relationship with chronic kidney disease, supported by the recommended references.
This information has been included in the introduction section of the new versión of the manuscript (Page 2, paragraph 3, and lines 61-62):
“Alterations in gut microbial homeostasis and the generation of metabolites associated with dysbiosis may contribute to kidney damage, ultimately promoting renal fibrosis. [doi: 10.1097/IMNA-D-23-00017 and 10.1111/joim.20089].”
Comments 9: The authors demonstrated that probiotic supplementation decreased inflammation and tended to decrease IS levels in the experimental model. However, did not lead to significant improvements in kidney function and VC neither rats nor patients. The latest several studies reported Targeting Lactobacillus johnsonii reverse chronic kidney disease, Probiotic interventions in peritoneal dialysis: A review of underlying mechanisms and therapeutic potentials.
Response 9: Thank you for bringing these recent studies to our attention and giving us the chance to put our findings into a broader context with the current literature.
Effectively, several reports, such as the study of Hua Miao, have shown encouraging kidney-protective effects of Lactobacillus-based probiotics. Hua Miao and colleagues gave Lactobacillus johnsonii (10⁹ CFU in 0.2 mL of sterile PBS) by oral gavage once a day for 21 days to rats with renal damage by an adenine-rich diet. They saw clear improvements in kidney health, less inflammation, and lower levels of toxins in the blood in a model of chronic renal failure rats induced by adenine with moderate injury and a relatively short timeframe.
In contrast with this study, our experimental model differed substantially in both design and severity. We employed the 5/6 nephrectomy (Nx) model, which induces a more advanced and irreversible form of renal damage and more importantly accompanied by chronic systemic inflammation and vascular calcification. Furthermore, we extended the probiotic supplementation period in an effort to capture long-term effects; however, the chronicity and advanced pathology of this model likely limited the ability of the probiotic to reverse structural and functional alterations. Although we observed a reduction in inflammatory markers and a tendency toward lower indoxyl sulfate levels, these effects did not translate into measurable improvements in renal function or vascular calcification. This suggests that, in advanced CKD, the inflammatory and oxidative milieu may override the potential benefits of our probiotic modulation.
We attribute this to the very advanced stage of CKD, inflammation, and vascular calcification from our experimental model. Future research should consider earlier intervention, longer treatment duration, and potentially specific formulations in order to optimize the probiotic effects in CKD.
We have included the following information in the discussion section of the manuscript (page 14, paragraph 2, and lines 546-555)
“Recent studies have reported kidney‐protective effects of specific Lactobacillus strains, such as Lactobacillus johnsonii, particularly in models with moderate renal damage and shorter disease evolution (10.1038/s41392-024-01913-1). In contrast, our study employed a 5/6 nephrectomy model characterized by advanced and irreversible renal injury, chronic systemic inflammation, and established vascular calcification. Although probiotic supplementation reduced inflammatory markers and tended to lower indoxyl sulfate levels, these effects did not translate into improvements in renal function or vascular calcification. This suggests that, in advanced CKD, the profound inflammatory and pro-calcifying milieu may limit the therapeutic impact of probiotic modulation. Future studies should evaluate interventions at earlier disease stages and explore targeted formulations to enhance clinical benefit.”
Comments 10: Please provide both manufacturer’s name and location (city, state, and country) for important equipment and reagents in the manuscript.
Response 10: Thank you for this comment. We have included this information in the new version of the manuscript.
Comments 11: The total concentrations of pCS, pCG, IS, and IAA showed statistically significant differences across different groups, but in Figure 4, bacterial diversity in the rat experimental model did not show a statistically significant difference. How can this result be explained?
Response 11: We appreciate the reviewer's comment. The fact that the bacteria responsible for producing these toxins constitute only a small fraction of the entire gut microbiota may explain the disparity between the significant changes in uremic toxin levels and the lack of statistical differences in bacterial diversity metrics. So, changes in the abundance of certain taxa with important metabolic functions may not always translate into variations in the alpha or beta diversity. This behavior had been observed previously as described in the review by Nadja B Kristensen and colleagues (Doi: 10.1186/s13073-016-0300-5), where they observed how probiotic interventions in healthy adults did not show consistent effects on global diversity indices, although they did show specific changes in taxa.
Taxonomic analysis in our study showed significant changes in the Ruminococcaceae family, which is related to the metabolism of uremic toxins. Without appreciably altering overall diversity metrics, these changes in composition may have functional effects on the metabolic output of the microbiota, such as the synthesis of phenolic and indole chemicals.
To improve our manuscript according to the reviewer comments, we have included in the discussion of the revised version of the manuscript the following text (page 16, paragraph 1, and lines 646-653):
“Although probiotic administration induced significant shifts in the Ruminococcaceae family, implicated in uremic toxin metabolism, these changes did not reflect global alterations in overall community diversity or composition, but rather targeted functional modifications within specific bacterial taxa. This behavior had been observed previously as described in the review by Nadja B Kristensen and colleagues (Doi: 10.1186/s13073-016-0300-5), where they observed how probiotic interventions in healthy adults did not show consistent effects on global diversity indices, although they did show specific changes in taxa.”
Comments 12: Previous studies show that these metabolites are associated with renal functions in serum or renal tissue of UUO and CKD patients, such as “A possible role of p-cresyl sulfate and indoxyl sulfate as biomarkers in the prediction of renal function according to the GFR (G) categories”, “Tryptophan metabolism as a target in gut microbiota, ageing and kidney disease”. The authors demonstrated that probiotic supplementation decreased the plasma concentrations of the total fraction of pCS, pCG, IS, and IAA in NX rats, while probiotic supplementation decreased the total concentrations of pCS, pCG, IS, and IAA. These results are paradoxical with previous publications. The reviewer suggests discussing and comparing this study with previous publications.
Response 12: We appreciate the reviewer's comments and the references provided. We have expanded the Discussion section to compare our findings with the cited literature and clarify why our observations are not necessarily paradoxical, but rather reflect different physiological contexts between experimental and clinical settings.
In our experimental model of 5/6 nephrectomy (Nx) in rats, probiotic supplementation resulted in a trend toward a reduction (p=0.05) in total plasma IS and a similar trend for pCS and pCG. This observation suggests that the probiotic intervention modulated gut microbiota metabolism, decreasing the production of indolic and phenolic compounds derived from tryptophan and tyrosine catabolism. These findings are consistent with the mechanism described in previous studies, which propose microbial regulation of tryptophan metabolism as a therapeutic target in CKD (Miao H. 2025 10.7150/ijbs.115359; Agus A. 2018 10.1016/j.chom.2018.05.003; Roager H.M. 2018 10.1038/s41467-018-05470-4).
However, in our clinical study, which included only patients with stage 5 CKD, we did not observe significant changes in circulating uremic toxin concentrations after probiotic supplementation. This discrepancy may be explained by the pathophysiological differences between advanced CKD and the experimental model. In end-stage renal disease, renal clearance of protein-bound solutes is profoundly impaired, and even a substantial reduction in their microbial generation may not be sufficient to detectably decrease plasma concentrations (Lin C. 2015 10.1371/journal.pone.0132589). In contrast, Nx rats retain residual renal function and intestinal excretory capacity, allowing the decrease in microbial production to be reflected in plasma levels.
Furthermore, several published studies reporting strong associations between IS, pCS levels and renal dysfunction primarily involve patients with varying GFR categories, rather than exclusively stage 5 populations (Corradi V. 2024 10.1097/IMNA-D-24-00002). And, as you can see in the following Figure, our own studies (which are pending publication) also show this relationship.
Therefore, our findings in rats with residual function and in humans with end-stage renal disease represent distinct points along the same pathophysiological continuum: the probiotic effect on toxin production may still occur in patients, but its biochemical impact is masked by the severely reduced excretory function typical of stage 5 CKD.
We have incorporated these considerations in the revised Discussion (page 14, paragraph 4, and lines 567-574).
“Probiotic administration was effective in reducing uremic toxins in our experimental model of vascular calcification; however, it failed to do so in the clinical study. This discrepancy may be explained by the pathophysiological differences between advanced CKD and the experimental models. In end-stage renal disease, renal clearance of protein-bound solutes is profoundly impaired, and even a substantial reduction in their microbial generation may not be sufficient to measurably decrease their plasma concentrations (Corradi V. 2024 10.1097/IMNA-D-24-00002 ; Lin C., 2015, 10.1371/journal.pone.0132589). In contrast, Nx rats retain residual renal function and intestinal excretory capacity, allowing the reduction in microbial production to be reflected in plasma levels.”
Comments 13: The authors demonstrated that probiotic supplementation decreased the plasma concentrations of the total fraction of pCS, pCG, IS, and IAA in NX rats, while probiotic supplementation decreased the total concentrations of pCS, pCG, IS, and IAA. The underlying molecular mechanism of this result is unclear. Reviewers suggested that the authors elucidate the potential molecular mechanism through in vitro and in vivo experiments. Alternatively, authors could discuss and speculate on potential molecular mechanisms based on previously publications “Intrarenal 1-methoxypyrene, an aryl hydrocarbon receptor agonist, mediates progressive tubulointerstitial fibrosis in mice”, “Lactobacillus species ameliorate membranous nephropathy through inhibiting aryl hydrocarbon receptor pathway via tryptophan‐produced indole metabolites”. For example, pCS, pCG, IS, and IAA are all endogenous ligands of aryl hydrocarbon receptors, and some studies have demonstrated that these metabolites, acting as ligands for aryl hydrocarbon receptors, activate aryl hydrocarbon receptor signaling, mediating kidney injury and renal fibrosis.
Response 13: In our study, we observed that probiotic supplementation in the experimental 5/6 nephrectomy model resulted in a trend toward decreased plasma levels of IS, with marginal statistical significance (p = 0.05), without finding differences in pCS, pCG, or IAA. In the clinical study, conducted in patients with stage 5 CKD, no significant changes were observed in the total or free fractions of these toxins (IS, pCS, pCG, or IAA) after the probiotic intervention.
We appreciate the reviewer's observation regarding the need to discuss the possible molecular mechanisms underlying these results. In this regard, several studies have described how tryptophan and p-cresyl-derived metabolites, such as IS, IAA, pCS, and pCG, act as endogenous ligands of the aromatic hydrocarbon receptor (AhR), whose activation has been linked to oxidative stress, inflammation, endothelial dysfunction, and progressive renal fibrosis (Cao et al., 2022 doi:10.1038/s41401-022-00914-6; Xie et al., 2024 doi:10.1186/s11658-024-00550-4; Sallée et al., 2014 doi: 10.3390/toxins6030934; Gondouin et al., 2013 doi: 10.1038/ki.2013.133). Therefore, reductions in these metabolites could imply indirect modulation of the microbiota–AhR–kidney axis, with potential pathogenic relevance.
Additionally, it has been reported that probiotics' effects on tryptophan metabolism extend beyond lowering IS or IAA levels; they can also encourage the production of other indoles with antagonistic or modulatory activity on the AhR, such as indole-3-propionic acid, which has anti-inflammatory and nephroprotective qualities (Miao et al., 2024 doi: 10.1111/bph.16219; Roager & Licht, 2018 doi: 10.1038/s41467-018-05470-4). Probiotics may therefore have a more qualitative than quantitative effect on the AhR pathway, influencing the kind of ligands generated rather than their overall concentration.
The following text explaining the potential molecular mechanism whereby probiotic modulate uremic toxins levels have been included in page 13, paragraph 6, and lines 528-536 .
“It is interesting to note that these metabolites (IS, IAA, pCS, and pCG) act as endogenous ligands of the aryl hydrocarbon receptor (AhR), which controls the fibrotic and inflammatory pathway in chronic kidney disease. By altering tryptophan metabolism, some Lactobacillus strains can reduce AhR agonist ligands and stimulate the production of beneficial indole derivatives with anti-inflammatory and antioxidant properties, such as indole-3-propionic acid (Cao et al., 2022 doi:10.1038/s41401-022-00914-6; Xie et al., 2024 doi:10.1186/s11658-024-00550-4; Miao et al., 2024 doi: 10.1111/bph.16219; Roager & Licht, 2018 doi: 10.1038/s41467-018-05470-4). Therefore, a qualitative alteration of AhR-related metabolites cannot be ruled out, although our data did not demonstrate a significant quantitative reduction of all uremic toxins. Future research should investigate this pathway, assessing AhR activation and the expression of its downstream targets in renal tissue.”
Comments 14: The clinical studies should be discussed and improved.
Response 14: We appreciate the reviewer's comment. We have revised the text to clarify that our clinical trial was designed as an exploratory pilot study to evaluate the potential effects of probiotic supplementation (Probimel) in patients with advanced chronic kidney disease (CKD). Given the small sample size and the advanced stage of disease in our participants (all in stage 5 CKD), the study lacked sufficient statistical power to detect clinically important differences, but rather to provide preliminary information on safety, feasibility, and potential biological effects, particularly on uremic toxin levels and vascular calcification.
We have modified the discussion to emphasize the exploratory nature of the clinical trial and highlight the limitations of its design, such as the sample size and the chronicity of the disease. However, we consider that the results provide valuable information to guide future, larger-scale randomized controlled trials evaluating the clinical efficacy of specific probiotic supplements in CKD populations.
The following text has been included in page 16, paragraph 2, lines 659-668:
“Our clinical trial was designed as an exploratory pilot study to evaluate the potential effects of probiotic supplementation (Probimel) in patients with advanced CKD. Given the small sample size and the advanced stage of disease in our participants (all in stage 5 CKD), the study lacked sufficient statistical power to detect clinically important differences, but rather to provide preliminary information on safety, feasibility, and potential biological effects, particularly on uremic toxin levels and VC. Although the study has a small sample size, the paired statistical analysis, in which each patient serves as their own control, allows us to conclude that under the specific conditions of our study, six months of probiotic supplementation does not modify renal function or the degree of vascular calcification “
Comments 15: The latest publications should be further discussed.
Response 15: We thank the reviewer for this valuable suggestion. We have carefully reviewed the Discussion section to incorporate and compare our findings with the most recent publications.
Comments 16: Please change the references based on the guide for authors.
Response 16: Thank you very much for your attention. We have updated the references to include more recent publications and have added them using Zotero to avoid errors.
Comments 17: The references 2-4, 37, 45, 71 are out of date. Please remove or cite the latest publications.
We thank the reviewer for pointing out the validity of several citations. We reviewed references 2-4, 37, 45, and 71 and took the following measures:
We have replaced the previous citations (2-4 and 37) with more recent and relevant publications. Reference 45 talks about a topic that I have not found in another paper, and reference 71 describes the CKD-EPI formula by which the eGFR is calculated. The updated references provide current evidence and better contextualize our results within the current literature. The manuscript and reference list have been updated accordingly.

Reviewer 3 Report (New Reviewer)
Comments and Suggestions for Authors
In the present manuscript titled “Effects of Probiotics on Kidney Function, Uremic Toxins, and Vascular Calcification in CKD: Evidence from Experimental and Human Studies” the authors carefully designed translational study combining experimental and clinical data to assess the effects of probiotic supplementation on kidney function, inflammation, uremic toxins, and vascular calcification (VC) in CKD. However, the overall novelty is moderate, as previous studies have addressed probiotic effects in CKD, though not always with such parallel animal–human comparison. The main limitation lies in the small sample sizes, particularly in the Nx+Prob rat group and clinical cohort, which likely reduced statistical power and may explain the absence of significant findings. Moreover, the discussion tends to overinterpret nonsignificant trends and correlations, which should be framed more cautiously as exploratory.
The authors are encouraged to clarify the rationale behind the chosen probiotic strain, its dosage, and treatment duration, and to discuss whether earlier intervention, longer treatment, or different formulations might yield stronger effects. The randomization and adherence procedures in the clinical study should be described more clearly.
The discussion could be streamlined to focus on the translational significance and limitations of the null results rather than extended mechanistic speculation.
- Abstract: “tented” should be“tended.”
- Use consistent abbreviations (VC, Nx, CKD) throughout the manuscript.
- Correct “interleuquin” to “interleukin” in all instances.
- Ensure figure legends include scale bars and consistent statistical notations.
- Provide CFU/day dosage and strain viability confirmation.
- Clarify if the clinical trial was registered.
- Simplify overly long sentences in the Discussion for readability.
- Add a color legend to Figure 6 correlation plots.
- Consider summarizing the clinical findings in a concise table in the Discussion.
Author Response
Thank you very much for taking the time to review this manuscript. Please find the detailed responses below and the corresponding corrections highlighted in red in the new version of the manuscript.
Reviewer 3
Comments 1:
In the present manuscript titled “Effects of Probiotics on Kidney Function, Uremic Toxins, and Vascular Calcification in CKD: Evidence from Experimental and Human Studies” the authors carefully designed translational study combining experimental and clinical data to assess the effects of probiotic supplementation on kidney function, inflammation, uremic toxins, and vascular calcification (VC) in CKD. However, the overall novelty is moderate, as previous studies have addressed probiotic effects in CKD, though not always with such parallel animal–human comparison. The main limitation lies in the small sample sizes, particularly in the Nx+Prob rat group and clinical cohort, which likely reduced statistical power and may explain the absence of significant findings. Moreover, the discussion tends to overinterpret nonsignificant trends and correlations, which should be framed more cautiously as exploratory.
Response 1: We acknowledge this comment and agree with the reviewer#3. In the new version of the manuscript we have pointed out in the text that the statistics of the experimental model were done with non-parametric models due to the small sample size and that the clinical trial is exploratory in nature. Furthermore, we have modified the discussion to avoid overemphasizing the trends.
Comments 2: The authors are encouraged to clarify the rationale behind the chosen probiotic strain, its dosage, and treatment duration, and to discuss whether earlier intervention, longer treatment, or different formulations might yield stronger effects. The randomization and adherence procedures in the clinical study should be described more clearly.
Response 2: We appreciate the reviewer’s comment regarding the selection of the probiotic used in this study.
The choice of this specific formulation was pragmatic: it was a commercially available probiotic preparation (Probimel) that contained bacterial strains (mainly from the Lactobacillus and Ruminococcaceae families). These strains have been previously reported by others in the literature to potentially modulate gut microbiota, reduce uremic toxins generation, and influence mineral metabolism (for Lactobacillus: Jens Walter, 2008, 10.1128/AEM.00753-08; Elaine Dempsey et al., 2022, 10.3389/fimmu.2022.840245, and for the Ruminococcaceae family: Q. Shang et al., 2016, 10.1039/C6FO00309E; Biddle A et al., 2013, 10.3390/d5030627; Jing Xie et al., 2022, 10.1002/mnfr.202100408).
The dose (109 CFU/mL) and the timing (45 days in rodents, 6 months in humans) were selected in accordance with trials available in the literature such as Mardhatillah Sariyanti et al., 2022 (10.18585/inabj.v14i4.2047) used in rats a dose of 1.5x10 8 or 1.5x10 9 CFU/mL/day during 21 days in rats. I-Kuan Wang et al., 2021 (10.3389/fnut.2021.661794) used to administrated 107-9 CFU/day during 42 days in rats and 2.5 × 109 CFU/day for 6 months in patients. Also, Na Tian et al., 2022, (10.3390/nu14194044) indicate that it is common to use 2 to 6 months in clinical trials. The same time as Shaun Sabico et al., 2019 (10.1016/j.clnu.2018.08.009) 2.5 × 109 CFU during 6 months.
This information has now been added to the new version of the manuscript page 13, paragraph 2, and lines 500-510:
“The probiotic used (Probimel) was chosen pragmatically as a commercially available product containing Lactobacillus and Ruminococcaceae. These strains have been previously reported by others in the literature to potentially modulate gut microbiota, reduce uremic toxins generation, and influence mineral metabolism [27–29]. The dose (109 CFU/mL) and the timing (45 days in rodents, 6 months in humans) were selected in ac-cordance with trials available in the literature, such as Mardhatillah Sariyanti et al., used in rats a dose of 1.5x108 or 1.5x109 CFU/mL/day during 21 days [30]. I-Kuan Wang et al. used to administer 107-9 CFU/day during 42 days in rats and 2.5 × 109 CFU/day for 6 months in patients [31]. Also, Na Tian et al. indicate that it is common to use 2 to 6 months in clinical trials [32]. The same time as Shaun Sabico et al. 2.5 × 109 CFU during 6 months [33]. “
Given that the study was exploratory in nature, our goal was not to test an optimized or customized probiotic design, but rather to evaluate whether supplementation with an available preparation could have measurable effects on kidney function, mineral metabolism, and vascular calcification in the setting of advanced CKD. We recognize that the efficacy of probiotics may vary substantially depending on the strains, combinations, and dosing regimens, and therefore, our findings should not be generalized to all probiotic formulations. Rather, they highlight the need for further research to define which microbial consortia, dosages, and treatment durations might be most effective in this context.
For the clinical study, randomization was performed automatically using the RedCap data collection system (Research Electronic Data Capture), aiming to homogenize the groups based on sex, age, glomerular filtration rate and albuminuria. In addition, all data were also collected in REDCap. This information was already initially provided as supplementary material along with the clinical trial flowchart (Supplementary figure 1), which graphically illustrates how, based on the inclusion and exclusion criteria, 23 patients were selected. After randomization, 14 were assigned to the VC group and 9 to the VC group receiving the probiotic. During the study, 3 patients were lost to follow-up from the VC group: two due to poor adherence and one who chose not to continue. Similarly, in the VC+Prob group, three patients were lost to follow-up: two due to kidney transplantation and one who decided not to continue in the study.
Comments 3: The discussion could be streamlined to focus on the translational significance and limitations of the null results rather than extended mechanistic speculation.
Response 3: According to the reviewer suggestion discussion has been divided.
Comments 4: Abstract: “tented” should be“tended.”
Response 4: Thank you for noticing this typographical error. We have already modified it in the abstract.
Comments 5: Use consistent abbreviations (VC, Nx, CKD) throughout the manuscript.
Response 5: We have carefully reviewed the entire manuscript and ensured that all abbreviations are used consistently and defined the first time they appear.
Comments 6: Correct “interleukin” to “interleukin” in all instances.
Response 6: Thank you for identifying this error. We have corrected all instances where it occurred.
Comments 7: Ensure figure legends include scale bars and consistent statistical notations.
Response 7: We appreciate this observation. All figure captions have been revised to include scale bars where applicable, and statistical annotations have been standardized across all figures.
Comments 8: Provide CFU/day dosage and strain viability confirmation.
Response 8: CFU/day dosage and strain have been included in the methods section of the revised manuscript (page 18, paragraph 2, lines 748-755).
“The probiotic selected was a commercially available formulation obtained from Probimel (Laboratorios Biotecnológicos Probimel S.L., Seville, Spain). It consisted of a liquid food supplement based on fermented milk (skimmed milk powder, 0.05 g, dissolved in 4.95 mL of water) containing 10⁹ colony-forming units (CFU)/ml of live probiotics, a mixture of species belonging to the Lactobacillus and Ruminococcaceae families. Specifically, the strain used was Lactobacillus acidophilus DMG017 (Probimel, Laboratorios Biotecno-lógicos Probimel S.L., Seville, Spain).”
Probimel is a commercially manufactured probiotic that is distributed with a Certificate of Analysis (CoA) that specifies batch-specific CFU counts and viability at the time of production. As the product is provided in its final, ready-to-use form, independent viability testing of the strain was not technically feasible within the scope of this study. However, to ensure the quality of the administered product, we relied on the manufacturer’s CoA and verified that all units used in the study were within the stated shelf-life and storage conditions recommended to maintain viability.
Comments 9: Clarify if the clinical trial was registered.
Response 9: Thank you for the appreciation. This clinical trial was registered in our institution. The name of the study was Mickid (Microbiota and kidney). This study was conducted under the Declaration of Helsinki, and the protocol was approved by the Ethics Committee of Reina Sofia University Hospital (Cordoba Research Ethics Committee, Spain; Record number: 4922, Committee file number: 332). You can see this information in the methods section (page 21, paragraph 1, and lines 869-872).
Also, the datasets generated during the current study are available in the European Nucleotide Archive (ENA) repository, Project accession: PRJEB87598. You can see this information in the Data Availability Statement (page 23, paragraph 7, and lines 984-985).
As shown in the screenshot, the clinical trial has been registered in ClinicalTrials, although it still needs to be reviewed by the platform administrators. Given the short 10-day revision period, there was not enough time to receive the trial registration number. If the manuscript is accepted, this information will be included at the proof stage.
Comment 10: Simplify overly long sentences in the Discussion for readability
Response 10: Thank you for this helpful stylistic suggestion. We have thoroughly revised the Discussion section to simplify sentence structure, improve flow, and increase overall readability.
Comments 11: Add a color legend to Figure 6 correlation plots.
Response 11: Thank you very much for your comment. We have added a color legend clarifying that blue refers to positive correlations and red to negative ones to the Figure 6.
Comments 12: Consider summarizing the clinical findings in a concise table in the Discussion.
Response 12: Thank you very much for your suggestion. We have created a graphical abstract summarizing all the findings.

Reviewer 4 Report (New Reviewer)
Comments and Suggestions for Authors
The manuscript investigates the impact of probiotic supplementation on vascular calcification (VC), kidney function, and uremic toxins in both preclinical and clinical settings. The topic is relevant and timely, and the study attempts to bridge mechanistic and translational insights. However, the current version suffers from several methodological and structural limitations that weaken the overall impact. The comments below aim to help the authors improve clarity, rigor, and interpretability.
Major Concerns
- The number of participants (n=23) is too small to draw reliable conclusions in the clinical study. The limited sample size significantly reduces statistical power and increases the risk of false-negative results. A power analysis or justification of sample size should be provided, and the authors should explicitly state that the study is exploratory in nature.
- The rat model lacks a positive control demonstrating regression or inhibition of VC. Including a known anti-calcification agent (e.g., SNF472, phosphate binders, or vitamin K2) would confirm that the model is responsive to therapeutic modulation. Without this, it is difficult to validate whether the calcification observed is pharmacologically targetable.
- Although both models were analyzed, major compositional and metabolic differences exist between human and rodent microbiota. These should be acknowledged more explicitly, as they limit translational applicability. The Discussion should cite comparative microbiome studies and temper claims of direct extrapolation.
- The 45-day treatment in rats and 6-month supplementation in patients may not be long enough to affect chronic processes like vascular calcification or dysbiosis reversal. The authors should justify these durations and discuss whether longer interventions might yield more measurable changes.
- The study focuses mainly on taxonomic profiling without evaluating functional aspects (e.g., short-chain fatty acid levels, metagenomic pathways). Integrating such analyses—or at least discussing their absence—would strengthen the mechanistic interpretation.
- The Nx+Probiotic group included only six animals, and large standard deviations were observed in biochemical parameters. The authors should use non-parametric statistics or show individual data points to improve transparency and interpretability.
- The Discussion is disproportionately long relative to the manuscript and repeats content already covered in the Results and Introduction. Condensing the section to emphasize key findings, mechanistic insight, and clinical relevance would improve readability and focus. For example, he Discussion merges preclinical, microbiota, and clinical findings without clear separation. Reorganizing it into structured subsections (e.g., “Preclinical Findings,” “Clinical Findings,” “Microbiota,” and “Limitations”) would enhance flow and comprehension. Moreover, portions of the Discussion restate general knowledge about phosphate metabolism and probiotics as phosphate binders. The authors should reduce redundancy and focus instead on how their results expand or contrast with previous evidence.
Minor Suggestions
- Clarify whether dialysis modality (HD vs. PD) influenced baseline microbiota composition or calcification indices.
- Provide the exact CFU composition and relative abundance of bacterial strains in the probiotic product.
- Figures 2 and 7 would benefit from showing individual data points.
- Consider summarizing the main findings in a graphical abstract or schematic.
Author Response
Thank you very much for taking the time to review this manuscript. Please find the detailed responses below and the corresponding corrections highlighted in red in the new version of the manuscript.
Reviewer 4
The manuscript investigates the impact of probiotic supplementation on vascular calcification (VC), kidney function, and uremic toxins in both preclinical and clinical settings. The topic is relevant and timely, and the study attempts to bridge mechanistic and translational insights. However, the current version suffers from several methodological and structural limitations that weaken the overall impact. The comments below aim to help the authors improve clarity, rigor, and interpretability.
Major Concerns
Comments 1: The number of participants (n=23) is too small to draw reliable conclusions in the clinical study. The limited sample size significantly reduces statistical power and increases the risk of false-negative results. A power analysis or justification of sample size should be provided, and the authors should explicitly state that the study is exploratory in nature.
Response 1: We appreciate the reviewer's comment and we agree. We have modified the text to emphasize that our clinical trial was designed as an exploratory pilot study to evaluate the potential effects of probiotic supplementation (Probimel) in patients with advanced chronic kidney disease (CKD). Given the small sample size and the advanced stage of disease in our participants (all in stage 5 CKD), the study lacked sufficient statistical power to detect clinically important differences, but rather to provide preliminary information on safety, feasibility, and potential biological effects, particularly on uremic toxin levels and vascular calcification.
We have modified the discussion to emphasize the exploratory nature of the clinical trial and highlight the limitations of its design, such as the sample size and the chronicity of the disease. However, the results provide valuable information to guide future, larger-scale randomized controlled trials evaluating the clinical efficacy of specific probiotic supplements in CKD populations.
We have included the next text in the discussion section page 16, paragraph 2, lines 659-668:
“our clinical trial was designed as an exploratory pilot study to evaluate the potential effects of probiotic supplementation (Probimel) in patients with advanced CKD. Given the small sample size and the advanced stage of disease in our participants (all in stage 5 CKD), the study lacked sufficient statistical power to detect clinically important differences, but rather to provide preliminary information on safety, feasibility, and potential biological effects, particularly on uremic toxin levels and VC. Although the study has a small sample size, the paired statistical analysis, in which each patient serves as their own control, allows us to conclude that under the specific conditions of our study, six months of probiotic supplementation does not modify renal function or the degree of vascular calcification.”
Comments 2: The rat model lacks a positive control demonstrating regression or inhibition of VC. Including a known anti-calcification agent (e.g., SNF472, phosphate binders, or vitamin K2) would confirm that the model is responsive to therapeutic modulation. Without this, it is difficult to validate whether the calcification observed is pharmacologically targetable.
Response 2: Our research group has extensive experience in establishing experimental models of vascular calcification in rats. We develope two distinct models of vascular calcification: one corresponding to acute kidney injury (two weeks), characterized by severe vascular calcification induced by 5/6 nephrectomy (Nx5/6), a high-phosphate diet, and vitamin D; and a second, more chronic model lasting 45 days, which induces milder kidney damage and a lower degree of vascular calcification. The graph below compares both models in terms of kidney injury, phosphate levels and absorption, and the degree of vascular calcification.
Figure 1. Comparison of renal function, phosphate levels, and the degree of vascular calcification between acute and chronic models of renal injury–induced vascular calcification.
As mentioned, our group has also substantial experience testing different therapies aimed at reducing vascular calcification in the 14-day acute injury model. Using this model, we have previously demonstrated that treatments such as vitamin E (doi:10.1152/ajprenal.00355.2013), magnesium-enriched diets (10.1016/j.kint.2018.12.015) or calcimimetics (doi: 10.1038/ki.2008.546) can effectively reduce vascular calcification. Therefore, we consider that the 45-day model used in this study, with reduced kidney injury and milder hyperphosphatemia, is likewise suitable for investigating interventions to attenuate vascular calcification.
For this reason, we believe that the probiotic might represent a pharmacologically targetable approach and may exert beneficial effects on vascular calcification and kidney function.
Comments 3: Although both models were analyzed, major compositional and metabolic differences exist between human and rodent microbiota. These should be acknowledged more explicitly, as they limit translational applicability. The Discussion should cite comparative microbiome studies and temper claims of direct extrapolation.
Response 3: Thank you for this important point. We used the rat model to approximate and control probiotic exposure and to study microbiota–host interactions under highly standardized conditions. We acknowledge that rodents and humans have different baseline microbiota compositions and that colonization dynamics differ across species.
Moreover, Probiotics often act transiently (modulating metabolic activity and relative abundance rather than establishing permanent colonization), so absolute bacterial load in feces may not fully capture functional effects. In addition, the severe dysbiosis present in advanced CKD patients may be less amenable to modification than the dysbiosis induced experimentally in rats. Studies from our research group acknowledge the limited translational relevance of experimental models to clinical data; specifically, such models do not always accurately recapitulate the full spectrum of disease characteristics observed in humans (Cross-Species Molecular Similarities Based on Omics Approaches in CKD: Toward Improved Translation From Pre-Clinical Models to Humans. Doi:10.1002/pmic.202400136).
These caveats help explain why selective taxonomic shifts and biochemical signals observed in the rat model did not translate into clinical benefit in our patient cohort. Anyway, we observed that in our experimental and clinical conditions of vascular calcification and renal disease, probiotics supplementation did not exhibit significant positive effects on vascular calcification.
The following text has been included in the Limitations section (Page 16, paragraph 4, and lines 673-682) of the new version of the manuscript:
“It is important to highlight that, while rodent models, particularly rats, are invaluable for studying microbiota-host interactions under standardized and controlled conditions, significant differences exist between the basal microbiota composition and colonization dynamics of rodents and humans. These differences arise due to host-specific factors, including anatomy, genetics, dietary habits, and lifestyles. Comparative studies have shown that, although some phyla such as Firmicutes and Bacteroidetes dominate in both species, relative abundances, genus and species distributions, and metabolic profiles differ markedly [63]. Thus, while rat models offer mechanistic insight, direct extrapolation to human bacterial load and microbiome responses should be approached with caution [64]. “
Comments 4: The 45-day treatment in rats and 6-month supplementation in patients may not be long enough to affect chronic processes like vascular calcification or dysbiosis reversal. The authors should justify these durations and discuss whether longer interventions might yield more measurable changes.
Response 4: We appreciate the reviewer's comment and acknowledge the importance of justifying the intervention duration and discussing its potential impact on chronic processes such as vascular calcification and dysbiosis reversal.
Previous studies that showed measurable effects on the microbiota, inflammation, and arterial calcification at these intervals led to the selection of 45 days for rats and 6 months for humans. For example, Mardhatillah Sariyanti et al., 2022 (10.18585/inabj.v14i4.2047) treated the rats for 21 days. I-Kuan Wang et al., 2021 (10.3389/fnut.2021.661794) used to administer the probiotics during 42 days in rats and 6 months in patients. Shaun Sabico et al., 2019 (10.1016/j.clnu.2018.08.009) also treated the patients during 6 months. Also, Na Tian et al., 2022, (10.3390/nu14194044) indicate that it is common to use 2 to 6 months in clinical trials. However, all of them recognize that there is no universal standard and that longer studies may be required to fully evaluate effects on chronic processes like VC.
We agree that prolonged interventions could show more significant or additional effects in slow-developing processes such as VC. This idea has been mentioned into the potential limitations of the study in the new version of the manuscript (page 16, paragraph 4, and lines 683-692).
Comments 5: The study focuses mainly on taxonomic profiling without evaluating functional aspects (e.g., short-chain fatty acid levels, metagenomic pathways). Integrating such analyses—or at least discussing their absence—would strengthen the mechanistic interpretation.
Response 5: We thank the reviewer for this valuable comment. We fully agree that functional analyses, such as quantification of short-chain fatty acids (SCFAs) or metagenomic pathway prediction, would provide deeper mechanistic insight into the observed microbial changes.
During the research stay in which uremic toxins were quantified, we had also planned to measure SCFAs concentrations in fecal samples using UPLC. Unfortunately, due to a technical failure of the equipment, the quantification could not be completed, and therefore, SCFAs data were not available for this study.
Nevertheless, we have now included a discussion of this limitation in the revised manuscript. Specifically, we note that SCFAs such as acetate, propionate, and butyrate are key microbial metabolites that modulate inflammation, intestinal barrier integrity, and systemic oxidative stress, processes highly relevant in CKD and vascular calcification. The absence of functional data limits our ability to infer whether the taxonomic changes observed translated into meaningful metabolic shifts.
We have highlighted this as a limitation in the Discussion and proposed that future studies should combine taxonomic and functional profiling (page 16, paragraph 2, and lines 668-672)
“Our results would be much more significant with the quantification of SCFAs, such as acetate, propionate and butyrate; the absence of functional data limits our ability to infer whether the observed taxonomic changes translated into significant metabolic alterations, in future studies should combine taxonomic and functional profiling.”
Comments 6: The Nx+Probiotic group included only six animals, and large standard deviations were observed in biochemical parameters. The authors should use non-parametric statistics or show individual data points to improve transparency and interpretability.
Response 6: We thank the reviewer for their valuable observation. We agree that, given the small sample size in the Nx+Probiotic group and the variability in some biochemical parameters, nonparametric analyses are more appropriate. So, we are using a nonparametric statistical test, as shown in the revised Table 1. Also, to improve the transparency and interpretability of the data, we have modified Figure 2 to show the individual data for all experimental groups.
Comments 7: The Discussion is disproportionately long relative to the manuscript and repeats content already covered in the Results and Introduction. Condensing the section to emphasize key findings, mechanistic insight, and clinical relevance would improve readability and focus. For example, the Discussion merges preclinical, microbiota, and clinical findings without clear separation. Reorganizing it into structured subsections (e.g., “Preclinical Findings,” “Clinical Findings,” “Microbiota,” and “Limitations”) would enhance flow and comprehension. Moreover, portions of the Discussion restate general knowledge about phosphate metabolism and probiotics as phosphate binders. The authors should reduce redundancy and focus instead on how their results expand or contrast with previous evidence.
Response 7: Thank you very much for your feedback. We have modified and reduced the discussion, reorganizing it into structured subsections as the reviewer suggested although we have not included epigraph. Also, we focus on discussing our results with previous evidence.
Minor Suggestions
Comments 8: Clarify whether dialysis modality (HD vs. PD) influenced baseline microbiota composition or calcification indices.
Response 8: We conducted an exploratory comparative study and found no significant differences in baseline microbiota composition or vascular calcification indices between hemodialysis and peritoneal dialysis patients. However, due to the small number of patients in each subgroup, these observations should be interpreted with caution for this reason this information has been not included in the manuscript.
Comments 9: Provide the exact CFU composition and relative abundance of bacterial strains in the probiotic product.
Response 9: CFU/day dosage and strain have been included in the methods section of the revised manuscript (page 18, paragraph 2, lines 748-755).
“The probiotic selected was a commercially available formulation obtained from Probimel (Laboratorios Biotecnológicos Probimel S.L., Seville, Spain). It consisted of a liquid food supplement based on fermented milk (skimmed milk powder, 0.05 g, dissolved in 4.95 mL of water) containing 10⁹ colony-forming units (CFU)/ml of live probiotics, a mixture of species belonging to the Lactobacillus and Ruminococcaceae families. Specifically, the strain used was Lactobacillus acidophilus DMG017 (Probimel, Laboratorios Biotecno-lógicos Probimel S.L., Seville, Spain).”
Probimel is a commercially manufactured probiotic that is distributed with a Certificate of Analysis (CoA) that specifies batch-specific CFU counts and viability at the time of production. As the product is provided in its final ready-to-use form, independent viability testing of the strain was not technically feasible within the scope of this study. However, to ensure the quality of the administered product, we relied on the manufacturer’s CoA and verified that all units used in the study were within the stated shelf-life and storage conditions recommended to maintain viability.
Comments 10: Figures 2 and 7 would benefit from showing individual data points.
Response 10: We appreciate the suggestion and have modified figures 2 and 7, showing individual data points.
Figure 2:
Figure 7:
Comments 11: Consider summarizing the main findings in a graphical abstract or schematic.
Response 11: We fully agree with your comment. Following your suggestion, we have done a graphical abstract summarizing the main findings.

Round 2
Reviewer 2 Report (New Reviewer)
Comments and Suggestions for Authors
The authors have improved the manuscript. Therefore, I suggest that the manuscript is published.
Author Response
We sincerely appreciate this reviewer’s contribution to improving our work, and we are pleased with the decision to accept the manuscript for publication in Toxins.
Reviewer 4 Report (New Reviewer)
Comments and Suggestions for Authors
The authors have adequately addressed my previous comments.
For completeness, I would recommend only a minor wording adjustment. In the newly added sentence stating that probiotic supplementation “does not modify” renal function or vascular calcification, a slightly less definitive expression—such as “no clear evidence of improvement was observed”—would better reflect the exploratory design and the small sample size. This would prevent the statement from being interpreted as a conclusive negative finding.
Author Response
We fully understand the nuance highlighted by this reviewer and agree that it improves the interpretation of our results in light of the characteristics of our study. The sentence mentioned by the reviewer appears in both the abstract and the graphical abstract, and both have been revised accordingly. We sincerely appreciate the reviewer’s input, which has undoubtedly strengthened our manuscript.
This manuscript is a resubmission of an earlier submission. The following is a list of the peer review reports and author responses from that submission.
Round 1
Reviewer 1 Report
Comments and Suggestions for Authors
The article is interesting as it addresses the possibility of using probiotics, a simple and accessible alternative, in modulating challenging interventions in kidney disease, which could bring considerable benefits to patients. However, the study was poorly programmed and leaves something to be desired in some areas.
The number of participants is very low, the sample is very heterogeneous, patients at different stages of kidney disease, the probiotic used and the dose.
What are the reasons for choosing this probiotic, dosage and time of use?
Why carry out two parallel studies (clinical and experimental)?
What scientific benefit did this study bring?
How could probiotics improve kidney function?
What was the nutritional status of the individuals studied? Was there a change in eating patterns during this period?
Table 1 should have the baseline and final values for each parameter for each group.
Table 3 should have the baseline and final values for each parameter for each group.
Why were inflammation parameters not evaluated in the experimental study?
Were the inflammation parameters altered at the beginning of the clinical study?
Reviewer 2 Report
Comments and Suggestions for Authors
I have the following comments
- In introduction- the relation of dysbiosis and vascular calcification may be expanded further with more specificities than citing references
- Ideally methods section should come before results unless it is a journal policy
- Reason for increased Plasma Cr in NX probiotic group?
- Nature of probiotics is not explained-authors say it can not be disclosed which is not ideal
- Probiotics have not shown any change in bacterial biodiversity-Reason?
- How will rat model replicate human bacterial load?
Reviewer 3 Report
Comments and Suggestions for Authors
This manuscript addresses an important and timely topic—the effect of probiotic supplementation on kidney function, mineral metabolism, and vascular calcification in advanced CKD—through both animal and clinical studies. The dual approach combining experimental and human data is a clear strength of the paper. However, several aspects require clarification or improvement to enhance scientific rigor, transparency, and presentation.
Major comment:
- The current title is overly definitive and may overstate the generalizability of the findings. Given the specific experimental settings, strain composition, and small clinical sample size, a more neutral title would better reflect the scope and limitations of the study.
- The description of the probiotic product is vague. The manuscript should specify the commercial product name, supplier, bacterial species included, whether live or inactivated, formulation (e.g., capsule, powder), and known functionalities such as SCFA production. These details are essential for reproducibility and interpretation.
- While 16S rRNA sequencing is mentioned, essential methodological details are missing. The authors should specify the DNA extraction method, sequencing platform parameters, software used (e.g., QIIME2 or DADA2), ASV/OTU clustering criteria, and reference database (e.g., SILVA). These steps critically impact results and reproducibility.
- Although group randomization and labeling are described, the study does not clearly state whether blinding was applied during data collection or outcome assessment. This should be clarified to assess the risk of bias.
- The number of animals in the Nx+Prob group (n=6) is small compared to other groups, potentially limiting statistical power. The authors should justify the group sizes and address this limitation in the discussion.
Minor
- Data tables could be better aligned to enhance readability. Currently, there are inconsistencies in spacing and the annotation for p-values. The statistical notation (*, **, ***) should be unified across all figure legends (e.g., "*p < 0.05, **p < 0.01, ***p < 0.001 vs Sham group").
- In Figure 3, the statistical significance between groups is not clearly indicated. Actual p-values or appropriate asterisks should be added. In Figure 5, the y-axis uses several abbreviations for clinical parameters—please provide full names or a figure legend for clarity.
